

# OMI air-quality monitoring over the Middle East

Michael Barkley[1], Gonzalo Gonzalez Abad[2], Thomas P. Kurosu[3], Robert Spurr[4], Sara Torbatian[5], and Christophe Lerot[6]

[1]Earth Observation Science Group, Department of Physics and Astronomy, University of Leicester, UK.
[2]Atomic and Molecular Physics Division, Harvard-Smithsonian Center for Astrophysics, Cambridge, Massachusetts, USA.
[3]NASA Jet Propulsion Laboratory, Pasadena, California, USA.
[4]RT Solutions Inc, Cambridge, Massachusetts, USA.
[5]Air Quality Meteorologist, Air Quality Control Company (AQCC), Tehran, Iran.
[6]Belgian Institute for Space Aeronomy (BIRA-IASB), Brussels, Belgium.

*Correspondence to:* Michael Barkley (mpb14@le.ac.uk)

**Abstract.** Using Ozone Monitoring Instrument (OMI) trace gas vertical column observations of nitrogen dioxide ($NO_2$), formaldehyde (HCHO), sulphur dioxide ($SO_2$), and glyoxal (CHOCHO), we have conducted a robust and detailed time series analysis to assess changes in local air-quality for over 1000 locations (focussing on urban, oil refineries, oil ports, and power plant targets) over

the Middle East for 2005–2014. Apart from $NO_2$, which is highest over urban locations, average tropospheric column levels of these trace gases are highest over oil ports and refineries. The highest average pollution levels over urban settlements are typically in Bahrain, Kuwait, Qatar, and United Arab Emirates.

     We detect 278 statistically significant and real linear $NO_2$ trends in total. Over urban areas $NO_2$

increased by up to 12% $yr^{-1}$, with only two locations showing a decreasing trend. Over oil refineries, oil ports, and power plants, $NO_2$ increased by about 2–9 % $yr^{-1}$. For HCHO, 70 significant and real trends were detected, with HCHO increasing by 2–7 % $yr^{-1}$ over urban settlements and power plants, and by about 2–4 % $yr^{-1}$ over refineries and ports. Very few $SO_2$ trends were detected, which varied in direction and magnitude (23 increasing and 9 decreasing). Apart from two locations

where CHOCHO is decreasing, we find glyoxal tropospheric column levels are not changing over the Middle East. Hence for many locations in the Middle East, OMI observes a degradation in air-quality over 2005–2014. This study therefore demonstrates the capability of OMI to generate long-term air-quality monitoring at local scales over this region.

## 1 Introduction

It is well-established that poor air-quality can significantly impact human health, ecosystems and agriculture, the built environment, and regional climate (Monks et al., 2009). The human and monetary costs associated with increasing levels of air-pollution are substantial. For example, the World



Health Organisation (WHO) estimates that globally nearly 7 million premature deaths were attributed to household and ambient air pollution during 2012 (WHO, 2014). Similarly, the Organ-
isation for Economic Co-operation and Development (OCED) estimated that outdoor air pollution is costing its 34 member states, plus the People's Republic of China and India, an estimated 3.5 trillion dollars a year in terms of the value of lives lost and ill health (OCED, 2014).

Regulatory control of urban air-quality is typically most effective when important target areas are extensively monitored, so pollutant emissions can be quantified and their atmospheric chemical pro-
cessing well-understood. In addition to in situ measurements made by ground stations and aircraft campaigns, satellite observations also form an important component of air-quality monitoring (Martin, 2008; Duncan et al., 2014). Satellite measurements of pollutants such as nitrogen dioxide ($NO_2$), carbon monoxide (CO), troposphere ozone $O_3$, formaldehyde (HCHO), sulphur dioxide ($SO_2$), and aerosol particulate matter (PM), have been widely used to characterise their global atmospheric dis-
tributions, to quantify surface precursor emissions, and to evaluate local air-quality (see e.g., Martin, 2008; Streets et al., 2013; Duncan et al., 2014, and references therein). Furthermore, with the advent of successive and longer-duration satellite missions, particularly the sequence of ultra-violet–visible (UV–VIS) instruments of GOME (Burrows et al., 1999), SCIAMACHY (Bovensmann et al., 1999), and GOME-2 (Callies et al., 2006), together with the OMI sensor (Levelt et al., 2006), there is a
growing ability to study long-term changes in air-quality from space. Some notable studies that have used satellite trace gas measurements to examine air pollution trends include: Richter et al. (2005); van der A et al. (2006, 2008); Ghude et al. (2008); De Smedt et al. (2010, 2015); Russell et al. (2012); Schneider and van der A (2012); Hilboll et al. (2013); Jin and Holloway (2015); Krotkov et al. (2015); Lamsal et al. (2015); Lelieveld et al. (2015), and Duncan et al. (2016).

The Middle East is a region where long-term changes in air-quality have probably been less well studied, in comparison to Asia, Europe, and North America. Besides frequent dust-storms (Furman and Hadar, 2003), the region's air-quality is characterised by year-long high ozone levels (Lelieveld et al., 2009), with an observed summer maximum (Liu et al., 2009; Zanis et al., 2014). The high ozone levels are in part due to long-range transport, strong local emissions, and favourable condi-
tions for ozone photochemistry (Lelieveld et al., 2009). In-situ instruments frequently record high pollutant concentrations in urban areas which exceed recommended guidelines (e.g., Modarres and Dehkordi, 2005; Nasralla and Seroji, 2007; Abdul-Wahab, 2009; Munir et al., 2013; Rashki et al., 2013).

The severity and variability of the region's air pollution can be directly observed from space.
For example, several studies have reported appreciable trends in $NO_2$ vertical column over Middle Eastern cities prior to 2011, with increases of order 2–9 % $yr^{-1}$ found over Tehran, 4–5 % $yr^{-1}$ over Jeddah, 6–7 % $yr^{-1}$ over Riyadh, and 10–20 % $yr^{-1}$ over Baghdad, depending on the satellite instrument and the averaging period (van der A et al., 2008; Schneider and van der A, 2012; Hilboll et al., 2013). Similarly, De Smedt et al. (2010) also found an increasing trend of 1–3 % $yr^{-1}$ in



GOME and SCIAMACHY tropospheric HCHO columns over Tehran and Baghdad during 1997–2009, but a decrease of about 2 % yr$^{-1}$ over Riyadh.

    More recently, Lelieveld et al. (2015) examined annual changes in OMI $NO_2$ and $SO_2$ columns over the Middle East over 2005–2014, reporting that after increases in the period 2005–2010, there was a reduction in these gases, either due to new regulatory legislation, or due to falls in economic

output associated with regional conflicts and geopolitical controls. In particular, decreases in $NO_2$ tropospheric columns of order 40–50% were noted over Damascus and Aleppo since 2011, coinciding with the start of Syria's civil war. A similar regional-scale study by Krotkov et al. (2015), also observed that during 2005–2008 OMI $NO_2$ column increased by 20% but remained approximately constant thereafter, whereas OMI $SO_2$ columns dropped by 20% after 2010, only recovering to 2005

levels in 2014. A further OMI $NO_2$ column trend analysis over the region's major cities by Duncan et al. (2016) likewise reported decreases over Damascus and Aleppo, but of about 3–4 % yr$^{-1}$, with increases of about 2–6 % yr$^{-1}$ elsewhere.

    However, besides these valuable studies and to the best of our knowledge, long-term changes in local air-quality for many smaller Middle-Eastern cities and towns have not been reported. In this

study we aim to remedy this situation by determining the changes in air-pollution over local population centres and also oil/energy infrastructure from space using a decade's worth of observations from the OMI instrument. Our target areas are therefore (1) cities and towns, (2) large-scale oil refineries and ports, and (3) coal, gas and oil fuelled electricity generating power-plants.

    To track the air-quality over our specified targets, we use OMI tropospheric vertical column ob-

servations of $NO_2$, HCHO, glyoxal (CHOCHO), together with retrieved boundary layer column concentrations of $SO_2$. Although each of these reactive species has different sources and sinks, they are all established key indicators of anthropogenic emissions, active photochemistry, and air pollution. For example, the dominant sources of $NO_x$ (=NO+$NO_2$) in the troposphere are the combustion of fossil fuels, biomass burning, emissions from soil, and lightning. Boundary layer $SO_2$ is pre-

dominantly generated by the burning of sulphur laden fossil fuels and the refinement of sulphur ores; volcanic $SO_2$ emissions are typically injected high into the atmosphere well above the boundary layer. The chemical reactions of $NO_2$ and $SO_2$ lead to the formation of nitrate and sulphate aerosols, which contribute to $PM_{2.5}$ (particulate matter with diameters <2.5 $\mu$m), another critical pollutant (Kim et al., 2015). HCHO and CHOCHO are reaction products from the oxidation of an-

thropogenic, biogenic, and pyrogenic volatile organic compounds (VOCs); they can also be directly emitted from fires. Observed HCHO and CHOCHO distributions therefore contain the signature of underlying VOC emissions. Quantifying VOC emissions is important, as the oxidation of VOCs in the presence of high $NO_x$ and sunlight leads to the formation of tropospheric ozone, which is a major air pollutant contributing to photochemical smog, as well as a key greenhouse gas and atmospheric

oxidant (Monks et al., 2015). Apart from $SO_2$, which has lifetime of about one week, the trace gases





are also relatively short-lived (order of hours to a day), so that spatial displacements from emission sources are often small.

In this study, we have two broad goals: (1) to establish which locations have the highest pollution levels and (2) to use time series analysis in order to determine which locations have statistically significant trends and to quantify the trend magnitudes. The structure of this paper is as follows. In section 2 we introduce the OMI data products. In Section 3 we discuss how the OMI data is gridded and describe the time series analysis. We present our results in Section 4, with a discussion on their validity in Section 5. Finally, we conclude the paper in Section 6.

## 2 OMI Satellite Data

### 2.1 The Ozone Monitoring Instrument (OMI)

The Dutch-Finnish Ozone Monitoring Instrument (OMI) (Levelt et al., 2006), is a nadir-viewing UV-VIS 2-dimensional charged-couple device (CCD) spectrometer, launched on board NASA's Aura satellite in July 2004. OMI orbits the Earth in a Sun-synchronous polar orbit, crossing the Equator at 13:30 local time (LT) in its ascending mode. The instrument has a $114°$ field-of-view producing a 2600 km wide swath which contains 60 cross-track pixels that range in size from $14 \times 26$ km$^2$ at nadir to $28 \times 160$ km$^2$ at the swath edges. With these viewing geometry and orbital characteristics, OMI achieves global coverage daily (in nominal operational mode). However, from 2007 onwards, OMI's coverage has been considerably reduced due to problems with certain rows of its CDD detector (see Section 3.1 for further details).

In this work we use 10 years (2005–2014) of OMI vertical column observations of $NO_2$, HCHO, CHOCHO, and $SO_2$. A brief overview of these products is given below; explicit details of each trace gas retrieval are given in the cited references.

### 2.2 SAO Formaldehyde

The official NASA HCHO product is provided by the updated Smithsonian Astrophysical Observatory (SAO) retrieval, as described in González Abad et al. (2015). HCHO slant columns are retrieved through a direct non-linear least-squares fitting of spectral radiances within the interval 328.5–356.5 nm. The retrieval algorithm includes dynamic calibration of solar and radiance wavelengths, the use of a daily radiance reference spectra, an under-sampling correction, and computation of common-mode residual spectrum. The cross-sections of HCHO and other absorbers are fitted, together with a Ring effect correction, scaling and closure polynomials, and a spectral shift parameter. The retrieved slant columns are converted to vertical columns using air-mass factors (AMFs) taken from look-up tables pre-computed using the VLIDORT radiative transfer model (Spurr, 2008), which uses a priori HCHO profiles from a global $2.0° \times 2.5°$ GEOS-Chem chemistry transport model simulation (originally described in Bey et al., 2001). Effective cloud fraction and cloud-top pressure are taken from





the OMI $O_2$-$O_2$ cloud product (Acarreta et al., 2004), whilst the surface reflectivity for clear-sky
scenes, is extracted from the OMI mode Lambertian Equivalent Reflectivity (LER) dataset created
by Kleipool et al. (2008). A daily post-processing normalisation correction (a function of latitude
and detector row) is applied to reduce retrieval biases, minimise noise and reduce cross-track strip-
ing. Over our domain of interest (20–80°E, 10–50°N) we find the median uncertainty of a single
measurement is about 60–70%, with an inter-quartile range (IRQ) of about 175%.

### 2.3   DOMINO $NO_2$ Vertical Columns

Tropospheric $NO_2$ vertical columns are from the KNMI DOMINO v2 product (Boersma et al.,
2011a, b). The $NO_2$ slant columns are retrieved using the differential optical absorption spectro-
scopic (DOAS) technique, by fitting the absorption cross sections of $NO_2$, $O_3$, and $H_2O$, along with
a synthetic Ring spectrum, to observed reflectance spectra in the 405-465 nm interval (Boersma et al.,
2007). The retrieved total slant columns are then assimilated into the TM4 chemistry model (Den-
tener et al., 2003), to estimate and remove the stratospheric $NO_2$ component. Finally, tropospheric
vertical columns are then derived by applying altitude-resolved AMFs from a precomputed look-up
generated with the KNMI DAK radiative transfer model using TM4 $NO_2$ profiles from a 2.0°×3.0°
global simulation. Surface reflectivity and cloud-parameters are taken from Kleipool et al. (2008)
and Acarreta et al. (2004), respectively. The median uncertainty of an individual measurement over
our target region is about 60–70%, with an IQR of about 54%.

### 2.4   BIRA CHOCHO Vertical Columns

Glyoxal vertical columns are retrieved from OMI spectra using an updated DOAS retrieval origi-
nally devised for GOME-2 (Lerot et al., 2010). The retrieval uses a daily mean Earthshine Pacific
radiance spectrum, and applies a row-dependent wavelength calibration. An initial pre-fit of liquid
water optical depth (in 405–490 nm) is performed, before the absorption cross-sections of CHOCHO
and other interfering absorbers are fitted in the spectral window 435–490 nm to retrieve the slant
columns. Tropospheric vertical columns are obtained using pre-computed look-up tables of altitude
resolved AMFs, which uses a priori profiles over land from a global 2.0°×2.5° IMAGES simulation
(Müller and Stavrakou, 2005). Over the oceans a single oceanic profile is used for the AMFs, derived
from airborne MAX-DOAS measurements over the Pacific Ocean (Volkamer et al., 2015). Unlike
the retrievals of HCHO and $NO_2$, the cloud effects are not accounted for in the AMF calculation (via
the independent pixel approximation); instead only 'clear-sky' observations are retained by rejecting
scenes with effective cloud fractions >20%. The median uncertainty of an individual measurement
is about 50-60% with an IQR of about 250%.





### 2.5 NASA Sulphur Dioxide

To assess changes in anthropogenic $SO_2$ we use planetary boundary layer columns (NASA product OMSO2 v1.2.0) determined using a principal component analysis (PCA) retrieval that is sensitive
to surface concentrations (Li et al., 2013). The algorithm applies the PCA technique to OMI UV sun-normalised radiances (310.5–340 nm) over an $SO_2$-free region (in the Equatorial Pacific) to establish the principal components of the main physical and measurement spectral features that do not correspond to $SO_2$ absorption. A set of principal components and $SO_2$ radiance Jacobians (i.e. that describe the radiance sensitivities to changes in the $SO_2$ column), are then iteratively fitted
to observed OMI radiances to obtain the slant column density. An estimate of the boundary layer $SO_2$ is then calculated based on the assumptions of the vertical $SO_2$ distribution (Krotkov, 2014). The algorithm applies the PCA technique to each OMI detector row individually, and uses the VLI-DORT radiative transfer code to compute the $SO_2$ Jacobians, with RT model inputs based on a fixed atmospheric profile and climatological $SO_2$ profile over the summertime eastern US, the latter corre-
sponding to an effective AMF of 0.36 (Fioletov et al., 2016). Scenes with strong ozone absorption of >1500 Dobson Units (1 DU = $2.69 \times 10^{16}$ molecules $cm^{-2}$) are excluded due to spectral interference in the retrieval. For this study we only use $SO_2$ data from OMI rows 4–54 (0-based) and with cloud radiance fraction <0.3 (Krotkov, 2014). Although $SO_2$ over the Middle East is mostly unaffected by volcanic emissions over 2005–2014 (Krotkov et al., 2015), transient volcanic $SO_2$ enhancements
were removed if they exceeded a threshold of 5 DU (Fioletov et al., 2011). The estimated uncertainty of the $SO_2$ PBL is about 0.5 DU (Li et al., 2013; Krotkov, 2014).

## 3   Methods

### 3.1   Data Gridding

We monthly-average the OMI observations onto a high resolution $0.05° \times 0.05°$ grid using an area-
weighting tessellation algorithm (Spurr, 2003; Hewson et al., 2015). The gridding algorithm properly accounts for the areal proportions of grid cells underlying the satellite footprint and inversely weights each observation according to the measurement uncertainty and OMI ground pixel-size. Spatial zoom orbits are not included in our analyses, nor are scenes with >20% fractional cloud cover and solar zenith angles > 70°. We also follow the data recommendations provided with each prod-
uct to reject non-optimum observations. The quality of the level 1B radiance data from certain rows of OMI's CCD detector is known to be affected by blockage effects, wavelength shifts, and stray light originating from outside the nominal field of view. This dynamic behaviour is the well-known OMI row anomaly (http://www.knmi.nl/omi/research/product/rowanomaly-background.php), which impacts atmospheric retrievals from the affected rows. The temporal variability of the row anomaly
is also known to compromise the derivation of long-term trace gas trends (De Smedt et al., 2015).





For this reason, we follow a similar approach to De Smedt et al. (2015), by using the OMI XTrack-QualityFlags (Dutch Space, 2009) to construct a static mask based on the most affected rows at the end of 2013, to then discard row-anomaly observations over the entire 2005–2014 period. This quality filtering largely removes any statistically significant sampling trends in the generated monthly datasets, except for the SAO HCHO and DOMINO $NO_2$ data. Therefore, for these gases we further use the XTrackQualityFlags to identify affected observations not flagged by the static mask, whose measurement uncertainties we then increase by a factor of 1000, so that these observations are included in the analysis without affecting the monthly averaged fields (i.e. they are assigned a very low weight in the averaging). On average there are typically 20–35 samples per grid-cell per month. To reduce noise in the monthly gridded data we smooth the data with a $0.15° \times 0.15°$ Gaussian filter of $1$-$\sigma$ width. For the noisier CHOCHO fields, a $2$-$\sigma$ Gaussian is used. The spatial smoothing enhances localised 'hot-spots' and is preferable to averaging the data onto a coarser resolution grid where the atmospheric signatures of target features can be lost. The median uncertainties of the gridded data are about 4% for $NO_2$, 6% for HCHO, 27% for CHOCHO, and 40% for $SO_2$.

Figure 1 shows the 2005 annual distributions of each species over the broad Middle East region. Clearly visible in the $NO_2$, HCHO and CHOCHO maps are the major urban areas of Riyadh, Baghdad, Tehran, Jeddah, and pollution enhancements along the eastern and northern coasts of the Persian Gulf. Intense $SO_2$ hotspots are found over the Jeddah and Mecca region, Kharg Island (in the Gulf) and near Kerman (Iran), where the Sarcheshmeh Copper Complex smelter is found.

## 3.2 Time Series Construction

To determine the temporal variability of the OMI vertical column data over urban areas we use the Global Rural-Urban Mapping Project Version 1 (GRUMPv1) settlement points database (Balk et al., 2006; SEDAC, 2015) to identify the geolocation coordinates of 818 cities and towns, that have populations ranging from over 50 to nearly 7 million inhabitants (as determined for the year 2000). We then construct a 10-year time-series (of 120 months) by averaging those monthly-gridded data that lie within $\pm 2$ grid-cells of the city or town location, which corresponds to a radial distance of approximately 10 km from the urban centre. Visual inspection of these spatial masks overlaid onto GoogleEarth imagery shows this filtering criteria is well-suited to capture the extent of most urban areas (see e.g., Figure 2). For Baghdad, Riydah, and Tehran, which have larger urban spread, the average of $\pm 4$ grid-cells is used instead, consistent with an approximate 20 km radial distance. In addition to the trace gas vertical columns, we also construct coincident times series for the associated cloud fraction, cloud-top pressure (or height), AMFs (where appropriate), and number of grid-cell samples. This helps clarify whether any observed trends in the trace gases are real, by applying the same time-series analysis to all the retrieved parameters.

We use this same approach to examine air-quality variability over oil refineries, oil ports and power plants. The 2010 Oil Refining Survey (Kootungal, 2010) was used to identify 41 major



crude oil refineries, whereas the Global Energy Observatory (GEO) free online resource (http://globalenergyobservatory.org) was used to locate 18 oil ports and 155 power plants. Where different types of power plants (e.g., oil versus gas fuelled) were closely co-located, a single geolocation co-ordinate was used to mark that target. The geographical distribution of the selected targets is shown in Figure 3a, and as an example, Figure 3b shows the urban spatial filtering masks, applied to the observed 2005 annual OMI $NO_2$ distributions over northern Iran. The latter figure shows that this approach treats the larger cities, such as Tehran, as individual urban regions, with intention of re-solving trends of separate districts, and that in some cases the spatial masks can partially overlap for targets which are close to one another.

Considering all locations, we find that by applying a weighted average to the data within $\pm 2$ grid-cells of the target, the median uncertainties of the trace gas vertical column time series reduce to 0.37% for $NO_2$, 0.73% for HCHO, 1.63% for CHOCHO, and 1.82% for $SO_2$.

### 3.3 Time Series Analysis

Each time series consists of monthly mean OMI observations $y(t)$ over a given target site, where time $t$ is in fractional years. We analyse each time series individually for each location following a consistent procedure. First, we filter the time series data for outliers, rejecting observations that lie beyond 3 median absolute standard deviations (Leys et al., 2013). This helps to reduce difficulties in the analysis and interpretation of noisy vertical column data when encountered; the filtering is not required for AMFs, cloud-parameters or the number of samples. Second, we linearly interpolate across missing data points, applying a quality filter that if >20% of values are missing from the time series then the dataset is rejected. However, we find that on average less than 6% of points may be missing from any given time-series. Third, we then fit to the data a model function $F(t)$ consisting of a linear component plus a four-term harmonic Fourier series, defined as follows:

$$F(t) = \mu + \omega t + \sum_{n=1}^{4} \left[ A_n \cos(2\pi n t) + B_n \sin(2\pi n t) \right] \tag{1}$$

where $\mu$ is the mean value of the time series data at time $t = 0$, $\omega$ is the linear trend of the variable (per year), and the parameters $A_n$ and $B_n$ are the Fourier series coefficients that essentially model seasonal and inter-seasonal variability. Many other trend studies have fitted very similar functions (see e.g, van der A et al., 2006; Gardiner et al., 2008; De Smedt et al., 2010; Hilboll et al., 2013; Jin and Holloway, 2015). We fit this model to the data using a non-linear least squares Levenberg-Marquardt algorithm, which generates an estimate of the fit parameters $\mu$, $\omega$, $A_n$ and $B_n$ plus their uncertainties and covariance.

Fourth, we check that the trend $\omega$ is real, i.e. significant at the 95% confidence level. The generally accepted rule for determining whether a trend is statistically significant is $|\omega/\sigma_\omega| > 2$, provided the lag-one autocorrelation of the fit residual is small (Weatherhead et al., 1998; van der A et al., 2006). To determine the precision ($\sigma_\omega$) of the trend we follow the approach of Gardiner et al. (2008) by





bootstrap resampling (with replacement) the initial fit residuals to reconstruct the fitted function with representative noise. The model function $F(t)$ is then refitted to this data and the fit parameters recalculated. This process is repeated 2000 times to build a sampling distribution for each of the fit

coefficients, with the difference between the 2.5th and 97.5th percentiles representing the coefficient's associated $2\sigma$ uncertainty (De Smedt et al., 2010). The advantage of the bootstrap resampling method is that it enables non-normally distributed data to be treated robustly (Gardiner et al., 2008).

Fifth, we additionally filter the initial fit residuals with short and long-term filters, to derive several other quantities of interest. For this we follow closely the curve fitting routines adopted by

the NOAA's Earth System Earth System Research Laboratory (ERSL) Global Monitoring Division (GMD), which are based on the original study of Thoning et al. (1989) and are fully described online (at http://www.esrl.noaa.gov/gmd/ccgg/mbl/crvfit/crvfit.html). The general approach is to transform the residuals into the frequency domain using a Fast Fourier Transform (FFT), apply a low-pass filter function to the frequency data, then transform the filtered data to the time domain using an inverse

FFT. We use a low-pass frequency filter $H(f)$ of the form:

$$H(f) = \exp\left[-\ln 2 \times \left(\frac{f}{f_c}\right)^6\right] \qquad (2)$$

where $f$ is the frequency (cycle per year), and $f_c$ is the frequency response of the filter. We filter the initial fit residual twice, once with a short-term cut-off value for smoothing the data, and once with a long-term value to remove any remaining seasonal oscillation and to track inter-annual variability.

Once the residual has been filtered, several curves can be constructed:

1. A *smoothed function fit* $F_S(t)$ which is the function fit $F(t)$ plus the residual filtered using the short-term cutoff value. Its uncertainty is given by $\sigma_{F_S}^2 = \sigma_F^2 + \sigma_S^2$, where $\sigma_S^2$ is the uncertainty of the short-term filtered residual, and $\sigma_F$ is the uncertainty of the model function estimated from the covariance matrix of the fit parameters using error propagation.

2. A *long-term trend fit* $F_T(t)$, which is the linear component of $F(t)$ plus the residual filtered using the long-term filter. It represents the long-term trend with the seasonal cycle removed. Its uncertainty is $\sigma_{F_T}^2 = \sigma_F^2 + \sigma_L^2$, where $\sigma_L^2$ is the uncertainty of the long-term filtered residual.

3. A *de-trended seasonal cycle* $F_C(t)$ which is computed by subtracting the long-term trend fit from the smoothed function fit, that is, $F_C(t) = F_S(t) - F_L(t)$. It represents the annual

seasonal oscillation with any long-term trend removed. Its uncertainty is given by $\sigma_{F_C}^2 = \sigma_{F_S}^2 + \sigma_{F_T}^2$. The average seasonal amplitude is defined as the median peak to trough difference of $F_C(t)$.

4. A *growth rate curve* $F_G(t)$ which is the rate of change of the long-term trend $F_T(t)$, determined using a three-point (quadratic) Lagrangian interpolation to compute the first derivative.

Its uncertainty is given by $\sigma_G^2 = 2 \times \sigma_{F_T}^2$ since the derivative is approximately equal to the



taking the difference of two data points one year apart, and plotting this difference midway between the two points. The average growth rate $G$, is then simply the median value of $F_G(t)$.

The statistical uncertainty of the residual filters ($\sigma_S$ and $\sigma_L$) are calculated following Thoning et al. (1989), via:

$$\sigma_{\text{filter}}^2 = \sigma_{rsd}^2 \sum_{i=1}^{n_c} c_i^2 + 2 \sum_{j=1}^{n_c-1} \sum_{k=j+1}^{n_c} c_j\, c_k\, r_{(k-j)} \tag{3}$$

where $\sigma_{rsd}$ is the residual standard deviation and $r_{(k-j)}$ are the lags in a first-order auto-regressive process (defined as $r(k) = r(1)^k$ for $k = 1, 2, \dots$), and $n_c$ is the number of filter weights. The filter weights $c_i$ are the values of the impulse response functions (IRF) of the filter transfer functions, which are measure of filter responses to a delta function impulse in the time domain (Thoning et al.,
1989). We compute the filter uncertainties for the short- and long-term cut-off values individually, and apply them to calculate the uncertainties in the derived curves. In this study, we use short-term and long-term filters of 200 and 667 days, respectively. Sensitivity tests using filters of 100, 150, 500, and 720 days, indicate that these two filter values offer the best compromise for slightly higher correlations between the data and fitted curves, versus slightly smaller curve uncertainties.

Figure 4 shows an example of a time-series fit to observed $NO_2$ data over Dahuk in Iraq (43.00°N, 36.87°E, population: 65683), where a statistically significant large upward linear trend of $2.77\pm0.35 \times 10^{14}$ molecules $cm^{-2}$ $yr^{-1}$ is found. This corresponds to a linear growth of $12.23\pm1.54\%$ relative to the observed 2005–2014 median VCD. In this example, $|\omega/\sigma_\omega| = 7.9$ and the uncertainties of the trend ($F_T$) and smoothed curves ($F_S$) are about 5% and 7.5%, respectively. The median growth rate $G$
is $12.44\pm8.18\%$, whilst the mean seasonal amplitude is $1.78\pm0.32 \times 10^{15}$ molecules $cm^{-2}$ (about $79\pm14\%$ relative to the median column). A similar analysis of the coincident time series of the $NO_2$ AMF, cloud fraction, cloud-top pressure and number of samples, reveals no other significant trend. This indicates that the upward growth in $NO_2$ is not caused by a trend in any other retrieval parameter and is real at the 95% confidence level. Figures S1 to S3 show similar fits for HCHO, CHOCHO,
and $SO_2$, respectively.

Although no underlying trends have been reported in the OMI data (Boersma et al., 2011b; Li et al., 2013; González Abad et al., 2015), as a precaution we performed the time series analysis on gridded OMI data over the remote Pacific ocean (60N–60S, 90–170W). No statistically significant trends were found for any species.

**4 Results**

In this section we present our main results, with Tables 1– 4 providing a concise summary of the analysis. Tables S1 to S2 (both excel files) provide a more complete classification of the analysis, where the highest ranked median levels and absolute linear trends for each target category are tabulated. Only the top 50 ranked locations are given for urban and power plants categories.





### 4.1  Average Pollution Levels

We use the observed median vertical columns for each category (i.e. not the fitted $\mu$ in equation 1) to assess average pollution levels (see Table S1). The overall average median columns are $26.98 \times 10^{14}$ molecules cm$^{-2}$ for NO$_2$, $3.97 \times 10^{15}$ molecules cm$^{-2}$ for HCHO, $18.71 \times 10^{13}$ molecules cm$^{-2}$ for CHOCHO, and 0.21 DU for SO$_2$. Relative to the overall median column, the highest mean NO$_2$

values are found over urban areas and oil ports, which are about 5% and 15% higher than refineries and power plants, respectively. Whereas for HCHO, we find that highest average columns are found over ports, with values about 5% higher than those for refineries, 15% higher than urban areas, and 16% higher than power plants. The highest average CHOCHO levels are also found over ports, with refineries, power plants, and urban areas being about 5%, 16%, and 25% less, respectively.

For SO$_2$ the highest average values are found over ports and refineries, which are 57% higher than power plants, and 81% higher than urban regions. Hence the general conclusion we can make is that average trace gas levels tend to be highest over oil ports and refineries (with the exception of NO$_2$).

The interquartile range (IQR) of the observed median columns gives some indication of the spread of the trace gas levels within a given target category. For NO$_2$ the IQR varies between 68% for

urban areas , 59% for refineries, 45% for ports and 53% for power plants (as a percentage of the overall average median values). The higher IQR over urban areas may reflect the larger variability of underlying NO$_x$ emissions between different towns and cities. For HCHO, the IQR only varies between 13% for urban areas and ports, to 20% for refineries and 26% for power plants. The similar IQR values likely indicate a lower variation in HCHO sources and sinks over different locations. For

CHOCHO, the IQR varies between 25% for urban areas to 51% over refineries, with the IQR over ports and plants about 46–47%. For SO$_2$, the IQR is lowest over urban areas (42%), but increases substantially over ports (100%), power plants (114%) to a maximum over refineries (148%).

Figure S4 shows the statistical box-and-whisker plots of the observed median vertical columns over urban targets for each country. The highest average pollution levels are typically found in

Bahrain, Kuwait, Qatar, and the United Arab Emirates (UAE). For example, NO$_2$ columns over Bahrain, Kuwait, and UAE, are 52%, 66% and 53% above the median urban level of $28.13 \times 10^{14}$ molecules cm$^{-2}$, respectively. Similarly for SO$_2$, Bahrain is 246% , Kuwait 490%, Qatar 208%, and UAE 166% above the average urban level, whilst for CHOCHO, the corresponding values are Bahrain 33%, Kuwait 72%, Qatar 30%, and UAE 58%. Meanwhile for HCHO, only Qatar at 30%

is prominently above the average urban level, whereas Bahrain is 18%, Kuwait 8%, UAE 19%, and Yemen 15%. According to the World Data Bank (http://data.worldbank.org) 2014 gross domestic product per capita, Bahrain, Kuwait, Qatar, and UAE were ranked 39[th], 22[th], 2[nd], and 21[st], respectively. By comparison, Iraq and Iran were ranked only 109[th] and 118[th]. Thus, over this region we find the more economically developed countries tend to have the highest pollution levels.

Considering average NO$_2$ levels over individual urban locations, we find Kuwait has 25 settlements in the top 50 highest ranked places, whereas Saudi Arabia has 9 locations, Bahrain 5, Iran 5,





and UAE 5. The highest average $NO_2$ columns were found over Funtas (in Kuwait), with a median value of $63 \times 10^{14}$ molecules $cm^{-2}$. Funtas is one of several coastal settlements adjacent to the Mina Al-Ahmadi and Mina Abdullah refineries, which all have high average $NO_2$ levels. These two re-
fineries had median vertical column levels of $55.07 \times 10^{14}$ molecules $cm^{-2}$, which were the highest of all refineries. Similarly, the adjacent Mina Al-Ahmadi port had the highest columns over all ports (of $64.11 \times 10^{14}$ molecules $cm^{-2}$). Thus, this general area is a particularly intense $NO_2$ hotspot, as indicated in Figure 1, due to the close proximity of several emission sources. For comparison, the highest $NO_2$ columns over power plants were found at the Tarasht Shahid Firouzi power station near
Tehran, which had a median level of $74.18 \times 10^{14}$ molecules $cm^{-2}$.

For HCHO, Iran with 16 locations, Yemen 13, and Saudi Arabia 6, dominated the top-50 ranked urban settlements. The five locations with the highest median values were located in Yemen, along its western coast, with the highest columns found over al-Marawiah, which had an average value of $5.28 \times 10^{15}$ molecules $cm^{-2}$. Over refineries, the highest median HCHO columns of $4.86 \times 10^{15}$
molecules $cm^{-2}$ were detected at the Umm Said refinery in Qatar. In addition, 6 of the highest 10 ranked power plants were also located in Qatar, particularly, those near Doha and Masaieed. However, higher median column values of 4.99 and $5.03 \times 10^{15}$ molecules $cm^{-2}$ were found over the Bandar-e Khomeni port, and Petroshimi and Bandar Immam power plants, respectively. Both these sites are closely located to the Bandar Imam Petrochemical facility in south Iran, which may
indicate a potentially strong VOC source.

The top-50 ranked urban locations for $SO_2$ were dominated by Kuwait (26 locations) and Iran (13 locations). The highest level of 0.73 DU was found at Rafsanjan (Iran), which is closely located to the Sarcheshmeh copper mine and smelter facility. The coastal settlements adjacent near to the Shuaiba, Mina Al-Ahmadi and Mina Abdullah refineries (in Kuwait) again had high pollutant levels,
typically 0.65–0.68 DU. Similarly, the oil ports of Kharg Island (in the Gulf) and Jeddah terminal also have high levels of 0.85 DU and 0.76 DU. Over power plants, the highest average levels were found over Mobin Petroshimi open gas cycle turbine (OGCT) facility, and the Asalooyeh OCGT near Bushehr (Iran), with levels of 1.00 DU and 0.98 DU, respectively.

The highest average CHOCHO values over urban areas were found mostly in Iran (17 locations),
Saudi Arabia (13), and Kuwait (17). The cities of Mecca and Tehran have the largest average levels of 44.32 and $39.44 \times 10^{13}$ molecules $cm^{-2}$, respectively. The highest levels over refineries are found at Tehran ($34.87 \times 10^{13}$ molecules $cm^{-2}$) and the three main coastal Kuwait refineries of Shuaiba, Mina Al-Ahmadi and Mina Abdullah ($30.34–30.80 \times 10^{13}$ molecules $cm^{-2}$). The oil port of Mina Al-Ahmadi also has the highest median level of $30.48 \times 10^{13}$ molecules $cm^{-2}$. The Mecca OCGT
power plant recorded the highest level of $43.81 \times 10^{13}$ molecules $cm^{-2}$; other notable CHOCHO levels over power facilities were at Tarasht Shahid Firouzi and Besat (in Tehran) with columns of 42.72 and $39.85 \times 10^{13}$ molecules $cm^{-2}$, respectively.





### 4.2 Observed Maximum Pollution Levels

We find median maximum values range from 41–79% above the overall median VCD for $NO_2$ (and are highest over ports), 68–92% for HCHO (highest over ports), 118–135% for CHOCHO (highest over refineries), and 93–318% for $SO_2$ (highest over refineries). However, the actual maximum values observed can be substantially higher. For example, the three highest $NO_2$ columns were found at (1) the 247.5MW Oil Besat Thermal Power Plant in Tehran, Iran (189.85 $\times 10^{14}$ molecules $cm^{-2}$, 604% above overall median VCD), (2) the Tarasht Shahid Firouzi power plant also in Tehran (167.78 $\times 10^{14}$ molecules $cm^{-2}$, 522% higher), and (3) over Tehran itself (136.65$\times 10^{14}$ molecules $cm^{-2}$, 407% higher).

For HCHO, the highest three values are found over the Iranian towns of (1) Shahrkord (10.49$\times 10^{15}$ molecules $cm^{-2}$, 164% higher than median VCD), (2) Somehsara (9.89$\times 10^{15}$ molecules $cm^{-2}$, 149%), and (3) Fuman (9.83$\times 10^{15}$ molecules $cm^{-2}$, 148%). Whilst for CHOCHO, the highest values are found in the towns of (1) Piranshahr (Iran) which had a maximum of 112.10$\times 10^{13}$ molecules $cm^{-2}$ (499%), (2) Tafileh in Jordan (96.57$\times 10^{13}$ molecules $cm^{-2}$, 416%), and (3) Qaemshahr in Iran (80.52$\times 10^{13}$ molecules $cm^{-2}$, 330%).

Lastly, for $SO_2$ the the three highest values were at the (1 & 2) Mobin Petroshimi OCGT CHP and Asalooyeh OCGT power plants, both in Bushehr (Iran), which had values of 2.22 DU (957%) and 2.05 (876%) respectively, and (3) the oil port at Kharg Island, which had a value of 1.98 DU (843%).

### 4.3 Seasonal Variability

For $NO_2$ median seasonal amplitudes are very similar irrespective of target categories. Typical amplitudes of 10–12 $\times 10^{14}$ molecules $cm^{-2}$ are observed (about 39-46% relative to the overall median). The highest seasonal amplitude of 66 $\times 10^{14}$ molecules $cm^{-2}$ was found over Besat Thermal Power Plant in Iran, whereas the lowest amplitude of 1.49 $\times 10^{14}$ molecules $cm^{-2}$ was determined over Ataq in Yemen. Unlike $NO_2$, HCHO exhibits greater seasonal variability (relative to the overall median) with seasonal amplitudes that vary from 3.02–3.43 $\times 10^{15}$ molecules $cm^{-2}$ (or about 60–86%) with the highest of 5.65 $\times 10^{15}$ molecules $cm^{-2}$ over Dahuk in Iran, and a lowest of 1.20 $\times 10^{15}$ molecules $cm^{-2}$ over Salalah OCGT power plant in Dhofar (Oman). Similarly, CHOCHO also exhibits large seasonal variability as median amplitudes vary from 14.38–16.48 $\times 10^{13}$ molecules $cm^{-2}$, or 77–88%, with the highest amplitude of 44.53 $\times 10^{13}$ molecules $cm^{-2}$ over Qaemshahr (Iran) and lowest amplitude over Bilin (Palestine) of 8.89 $\times 10^{13}$ molecules $cm^{-2}$. Even larger seasonal variability is demonstrated for $SO_2$, as amplitudes range from 0.21–0.44 DU (100–210%) with the lowest amplitude of 0.09 DU over Al Gaydah (Yemen) and highest over 1.15 DU over Shahreza (Iran). However, the higher amplitudes in the CHOCHO and $SO_2$ data may reflect the greater noise in the time series data compared to the other gases (see e.g., Figure S2–3).



### 4.4 Linear Trends $\mu$

For $NO_2$, statistically significant real linear trends were determined for 198 of 818 urban locations (a
detection rate of 24%). However, if we disregard Palestine, where only 1 out of its 274 urban targets
had a real trend, then this value increase to 36%. The corresponding 'detection percentages' for oil
refineries, oil ports, and power plants are 42%, 33%, and 37% respectively (Table 1). Urban trends
ranged from $-3.05\pm0.93$ to $12.23\pm1.54$ % per year (relative to the observed median column), with
the highest trend in Dahuk (Section 3.3, Figure 4). Generally, Iraq and Iran have the highest linear
$NO_2$ trends. For example, 5 of top 7 highest linear trends were found in Iraq , whereas 21 Iranian
cities appeared in the top 50 (see Table S2). Overall, the median linear trend is about 3% $yr^{-1}$,
although for the top 50 highest ranked locations, the trends were of order 2–12 % $yr^{-1}$, relative to
each locations observed median VCD. Only two locations showed a decrease in $NO_2$: Aleppo (a.k.a
Halab; of -2.43$\pm$1.06 % $yr^{-1}$) and As-Safira (-3.05$\pm$0.93 % $yr^{-1}$), both in Syria. On a per-country
basis, 75% of urban targets in Iraq had a real increasing trend, 58% in Iran, 53% Lebanon, 50%
Qatar, 44% Oman, 38% Saudai Arabia, 31% Jordan, 22% Syria, 11% Kuwait, 4% in Israel, 3%
Yemen, and Palestine <1%. No trends were detected in the UAE or Bahrain.

Trends over refineries ranged from about 2–6 % $yr^{-1}$, with a maximum found over the Daura
refinery, near Baghdad; its capacity is smaller than some of the larger refineries which may indicate
the influence of other nearby $NO_2$ sources. For Iran, Iraq, and UAE, about 60% of the refineries
studied showed an $NO_2$ increase. Trends over oil ports ranged from about 2–9 % $yr^{-1}$, with highest
trend found over Umm Qasr in Iraq. Trends over power plants ranged from about 2–8 % $yr^{-1}$, with
5 of top highest 10 trends found in Iran. The highest trends for power stations were detected over
the two Sabiya plants in Kuwait which had trends of 8.13$\pm$0.14 % $yr^{-1}$. Figure 5, which shows
the geographical distribution of $NO_2$ trends (top left panel), indicates that there are several local
and regional locations where consistent increases $NO_2$ are observed. For example, the are several
targets with increasing trends situated close to Riyadh, Tehran, Baghdad, Muscat (in Oman), as well
as Isfahan and Yazid (both in Iran). Regional trend hot-spots include the areas south-west of Tehran,
and a thin corridor stretching from northern Jordan to Lebanon, passing through south-west Syria.

For HCHO, statistically significant real linear trends were determined for only 4% of urban loca-
tions. The corresponding trend detection percentages are 15% for oil refineries, 22% for oil ports,
and 17% for power plants (Table 2). Urban trends ranged from 2–7 % $yr^{-1}$. The highest absolute
linear urban trend was found al-Wakrah in Qatar (Figure S1), although this was only about 5% $yr^{-1}$
in relative terms; the highest percentage trend was found over Attaif in Saudi Arabia (6.95$\pm$2.42 %
$yr^{-1}$; see Table S2). Generally, Saudi Arabia has the highest number of settlements with increasing
HCHO trends (13 of the corresponding 34 trends found). Elsewhere there were 6 trends found in
Oman, 5 in Qatar, 3 in Iran, 3 in UAE, 1 Iraq and 1 Israel. Only six trends were detected over oil
refineries, which ranged from 2–3.5 % $yr^{-1}$, with a maximum found over the Ruwais refinery in
UAE. Only four trends over oil ports were found, which ranged from 2–4 % $yr^{-1}$, with highest trend




found over the Ras Laffan port at Al Khawr (Qatar). Trends over power plants ranged from 2–7 % yr$^{-1}$, but were only found over Saudi Arabia (8 stations), Iran (5), Qatar (4), and UAE (5). Figure 5 (top right panel) shows that the target locations with trends are mostly found along the western Gulf coast, particularly along the Saudi Arabian coast near Ad-Dammam, and also near Doha in Qatar.

For SO$_2$, very few trends were detected, only 2% over urban targets, 7% over refineries, 11% over ports and 6% over power plants (Table 4). Over urban areas, trends ranged from −61.64±28.73 % yr$^{-1}$ in Hamismusayt (Saudi Arabia) to 118.49±42.78 in Azrashahr (Iran); the latter value is inflated by a median SO$_2$ column of approximately zero over its location. Notably, 11 of the 18 trends were detected in Iran. Over refineries, three trends were detected of about 9–15 % yr$^{-1}$. Over ports, the Iranian ports of Bandar-e Khomeni and Kharg Island, showed decreases of about 6% yr$^{-1}$. Lastly, over power plants 5 locations had decreasing trends of −7 to −22 % yr$^{-1}$, and 4 stations increasing trends of about 6–305 % yr$^{-1}$ (again the latter value has a median SO$_2$ column of approximately zero). Figure 5 (bottom left panel) shows the geographical distribution of the SO$_2$ trends; it is evident that several closely located targets exhibit similar trends, particularly near (1) Abba (in southwest Saudi Arabia), (2) around Neka, Sari and Behhshahr in northern Iran, which are close to the Shamid Salimi power plant at Nowzarabad, and (3) Tarbiz, Azrashahr, and Maragheh in Iran.

For CHOCHO only two trends were detected: (1) at Al-Hawr (Qatar) of −4.58 % yr$^{-1}$, and (2) at the Ras Laffan Power plant in Qatar of −4.83 % yr$^{-1}$, which are located close to each other (Figure 5; bottom right panel).

### 4.5 Median Growth Rates $G$

Figures 6 show the growth rates that correspond to sites with statistically significant linear trends (i.e. where we can be sure that the trend is not attributed to variations in other retrieval parameters), and Figure 7 shows their difference, here defined as the linear trend (in % yr$^{-1}$) minus the growth rate (in % yr$^{-1}$). While the overall median differences are quite small (typically less than 2 % yr$^{-1}$), for individual locations they can be much larger. Typically for NO$_2$ and HCHO, the differences can range between ±5 % yr$^{-1}$ (Table S3). Such occurrences are non-negligible. For example, the NO$_2$ linear trend and growth rates over Ibri (in Oman), were about 3 % yr$^{-1}$ and 8 % yr$^{-1}$, respectively. For SO$_2$ the differences can be even more substantial (i.e. tens of percent per year), but this is likely an effect from filtering much noisier fit residuals (e.g., as shown in Figure S3). Furthermore, Figure 7 show that there is no clear spatial patterns or coherence to the geographical distribution of the linear and growth differences. This raises the interesting question for AQ trend studies: should one consider linear changes in a trace gas, or use a growth rate? The former does not capture shorter-term variations in growth, but the latter is more susceptible to the choice of fit residual filter and has larger uncertainties.





## 5 Trends: Fact or fiction?

It is, perhaps, tempting to try explain the trends in terms of variations in underlying emissions and/or atmospheric chemistry. However, it seems more prudent to carefully assess the validity of our presented results instead. We do this by examining our analytical uncertainties, by performing a series of sensitivity tests, and by discussing our results in the context of other recent satellite studies.

### 5.1 Uncertainties

Calculated uncertainties in the linear trend and other derived curves mostly reflect the noise of the trace gases time series. For $NO_2$, and considering only those locations with real statistically significant trends, we find the median uncertainty in the linear trend is about 1%. The median uncertainties are about 4% for the trend curves, 6–7% for the smoothed curves, 5–6% in the growth curves, and about 6–8% seasonal-cycles, respectively. For HCHO, corresponding values are very similar; 5% in
the trend curves, 7% in the smoothed, 7% in the growth and 9% in the seasonal curves, with the median uncertainty in the linear trend about 1%. For CHOCHO, uncertainties are slightly higher, being 10% in the trend curves, 15% in the smoothed, 14% in the growth and 18% in the seasonal curves. The median uncertainty in the linear trend is about 2%. The uncertainties are noticeably higher for $SO_2$, with errors of 10–23% in the trend curves, 15–34% in the smoothed, 14–32% in the growth
and 18–40% in the seasonal curves. The median uncertainty in the $SO_2$ linear trend is about 2–5%. Thus for the most part, the uncertainties in the linear trends are comparable or smaller than than the trends themselves. However, uncertainties in the growth rates are comparable or higher than the derived average growth trends.

### 5.2 Sensitivity Tests

Subtle differences in the data analysis approach may affect the choice of locations where trends are detected, or may change existing trend directions and magnitudes. Given this situation several additional tests were carried out to determine the sensitivity of our results to various parameters. These are outlined below. Tables S4–S7 record how many trends were now detected in each test, and whether or not they were previously detected in the default approach (as outlined in Section 3.3).

– **Test 1: Construct each 10-year time series using a mask of $\pm 4$ grid-cells ($\sim$20 km radius around each target), instead of the default $\pm 2$ grid-cells ($\sim$10 km radius around each target).**

For $NO_2$, there were 22 and 12 extra trends detected over urban and power plants, respectively, compared to our default analysis (shown in Table 1). The number of $NO_2$ trends detected over
545 oil ports and refineries were unchanged. However, some trends previously detected were now missing, compensated by new trends over other locations. For example, in the case of urban targets, 170 locations in this test were also previously detected in the default scenario, but 28



of the previous trends were missing, and 50 new trends detected (see Table S34). Nevertheless, median linear trends were of similar order (about $-2$ to $10\ \%\ \mathrm{yr}^{-1}$). The highest urban $NO_2$

trends were still at Dahuk and Irbil (in Iraq), but marginally changed from $12.23\pm1.54\ \%\ \mathrm{yr}^{-1}$ to $10.39\pm1.42\ \%\ \mathrm{yr}^{-1}$, and from $8.76\pm1.17\ \%\ \mathrm{yr}^{-1}$ to $8.18\pm1.16\ \%\ \mathrm{yr}^{-1}$, respectively. Umm Qasr and Zubayr oil terminals were still the highest ranked ports, at $8$–$9\ \%\ \mathrm{yr}^{-1}$. Similarly, the Sabiya CGGT and OCGT power plants still had the highest trend of about $8\ \%\ \mathrm{yr}^{-1}$.

For HCHO this resulted in an extra 25 trends being detected (spread over all categories); never-

theless the median linear trends were still about $2$–$3\ \%\ \mathrm{yr}^{-1}$. The majority of trends previously found in the default case were still present in this test. However, notably, the number of detections over ports doubled from 4 to 8, with the highest trend now found over Sitrah in Bahrain $4.03\pm1.14\%$ (the trends over the Ras Laffan and Al-Ruwais ports were now displaced to second and third highest). That said, both al-Wakarah and Doha still had the highest (absolute)

trends for cities (of about $3$–$4\ \%\ \mathrm{yr}^{-1}$), and again the Ras Abu Fountas OCGT power plant in Doha (Qatar) also topped the highest ranked trends over power stations.

For $SO_2$ the highest urban trends were still over Ilam and Tabriz (Iran), and also Baghdad and Irbil (Iraq). Although 18 urban trends were still detected, only 9 were previously found in the default scenario (including those over Sari, Neka and Maragheh), thus 9 new trend

locations were found. Over ports the highest trend was now over Umm Qasr relative trend of $23.09\pm5.87\ \%\ \mathrm{yr}^{-1}$, as a trend was now not detected at Bandar-e Khomeni (BIK); Kharg Island still reported a decrease of about $-6\ \%\ \mathrm{yr}^{-1}$. $SO_2$ trends were found over the same three refineries, and were still about $9$–$12\ \%\ \mathrm{yr}^{-1}$. Median trends over urban areas changed from 9.8 to $12.0\ \%\ \mathrm{yr}^{-1}$, but refineries and power plants remained at about 9 and $6\ \%\ \mathrm{yr}^{-1}$,

respectively. The Sabiya CGGT and OCGT power plants in Kuwait were still the highest-ranked trends.

For CHOCHO, in addition to the trend of al-Hawr, only an extra urban trend at Shushtar (Iran) of $-5.33\pm2.58\ \%\ \mathrm{yr}^{-1}$ was detected. An additional trend at the Ras Laffan (Qatar) refinery was also found of about $-4.27\pm1.91\ \%\ \mathrm{yr}^{-1}$; the trend over the Ras Laffan power stations was

also present as in the default scenario.

In summary, although increasing the averaging radius to $\sim$20 km around each target has a small impact on overall median trends, it can result in differences at individual locations. Generally however, most of the trends found in the default approach were still present.

– **Test 2: Use of different cloud fraction filters (40% instead of 20%)**

For this we used only OMI $NO_2$ and HCHO observations with a cloud fraction $\leq$40% in the analysis; CHOCHO and $SO_2$ and were untested, as the use of a strict cloud threshold is advised (Krotkov, 2014).




For $NO_2$, there were 45 fewer trends detected in total (mostly over urban locations, 165 compared to 198 previously). The majority of trends in the default analysis were still present - hence the relative median trends were still about 3–4 % $yr^{-1}$. Furthermore, the top ten urban ranked trends were also mostly unchanged, as were those over ports, refineries and power plants compared to the default scenario.

For HCHO there were two less detections in total, with the highest urban trend found over Ardebil (Iran) at 7.06±2.53 % $yr^{-1}$. Again the majority of locations with trends were consistent with those detected in the default scenario; hence the median trends are still about 3 % $yr^{-1}$.

Thus, in summary, increasing the cloud fraction for $NO_2$ and HCHO slightly decreases the number of trends found, and on average, has a small impact on overall median trends. Differences at individual locations can still occur, compared to the default approach.

– **Test 3: Use of OMI detector rows totally unaffected by the row-anomaly (rows 5-23)**

Despite the use of the static masks and OMI level-1B XTrackQualityFlags in the data gridding (Section 3.1), statistically significant sampling trends are present for a number of locations where VCD trends occur (Tables 1–4). Therefore, we repeated our analysis but using only data from OMI's unaffected rows.

However, despite using data from the unaffected detector rows, we still find sampling trends present in some time series. For example, for $NO_2$ 227 statistically significant VCD trends were now detected in total, but 57 of those had a sampling trend. As a consequence, for $NO_2$ only 136 'real' trends in total were detected (instead of 278; see Table 1), in particular over 100 less trends were found over settlements, and nearly half the amount over power plants. Nevertheless, trends were found over the same locations (especially in the top 20 ranked urban) with median trends now 3.0–6.5 % $yr^{-1}$ (was 3–4 % $yr^{-1}$). The Daura refinery (near Baghdad) and the Umm Qasr port still had the highest trends of these categories (but now of 5.78±0.95 and 7.76±0.91 % $yr^{-1}$, respectively).

For HCHO, only 38 trends were detected in total (was previous total was 70), in particular, 27 urban and 16 power plant trends previously detected were now missing. Median linear trends about 3.0–5.0 % $yr^{-1}$, with the highest urban trends now found over over Sarepol (7.39±3.41 % $yr^{-1}$) and Torbatejam (8.41±2.60 % $yr^{-1}$), both in Iran.

For $SO_2$ there were 7 fewer trends detected in total, but notably 9 of the default of 18 urban trends were still detected. Median trends were about −7 to 13 % $yr^{-1}$ (formerly −7 to 10 % $yr^{-1}$). Urban trends now ranged from about −12 to 150 % $yr^{-1}$, and power plants −7 to 75 % $yr^{-1}$. A decrease of about 7 % $yr^{-1}$ was found over Kharg Island port, as was an increase of 9.69±2.40 % $yr^{-1}$ over the Daura refinery.





For CHOCHO, only two trends over the towns of Ardebil (of about 25 % yr$^{-1}$) and Shushtar (of about $-18$ % yr$^{-1}$), both in Iran, were detected.

Thus, in summary, we find that using less observations results in fewer trends being detected (since there are less observations in averaging which increases the noise). Also, we find that use of the unaffected rows results in slightly higher median trends overall.

**– Test 4: Increased smoothing of gridded monthly averages**

To further reduce noise in the gridded data we increase the degree of smoothing by using a larger Gaussian filter of $0.35° \times 0.35°$ with a 2-$\sigma$ width. This had a small effect on our results, as the total number of trends detected for all species and categories was more or less the same in the default scenario, and nearly all the trends locations in the default scenario are still detected. Median trends were also unchanged.

**– Test 5: No filtering for outliers in VCD time series analysis**

Whilst filtering for VCD outliers in the time series fitting is advantageous in improving the modelling fitting, it may also remove genuine data points that may affect trend detection.

For NO$_2$ the number of detections and their locations are approximately the same, albeit with 27 new urban trends replacing 26 default trends. Tehran is now the top-ranked city (in absolute terms) although its relative trend of 8.82$\pm$2.59 % yr$^{-1}$ is still less than that found at Dahuk (15.86$\pm$2.33 % yr$^{-1}$). Considering all target categories, we find median trends range from about $-2$ to 16 % yr$^{-1}$ (as opposed to about $-3$ to 12 % yr$^{-1}$). Similarly for HCHO, the number of detections and their locations are approximately the same, and the range of trends is unchanged.

For the most part SO$_2$ trends are also in the same places, with a few exceptions over urban and power plant targets. Trends over urban and refinery targets cover a higher range $-23$ to 148 % yr$^{-1}$, and 10–20 % yr$^{-1}$, respectively, than previously. This results in higher median values of 14 and 15 % yr$^{-1}$ for these categories (compared to the default case). Median values over ports and power plants are unchanged.

Lastly, for CHOCHO, only one urban trend was now detected over Shushtar (Iran) at $-8.65\pm3.83$ % yr$^{-1}$, and two extra trends were found over the closely located power plants both at the Az Zour plant complex in Mina Said (Kuwait), which had a trend of $-3.64\pm1.75$ % yr$^{-1}$.

In summary, not filtering for VCD outliers only has a small effect on our results. Typically, most trends in the default scenario are detected, albeit with some changes in trend magnitudes (mostly for NO$_2$ and SO$_2$).

**– Test 6: Focus only on cities with >500,000 people using a spatial mask of $\pm$16 grid-cells ($\sim$80 km radius around each target)**





In this test we focused on the integrated signals from large population centres, which correspond to 32 targets only. For $NO_2$ there are 22 trends detected (a detection rate of 69%), for HCHO there were 9 trends (32% detection rate), and $SO_2$ 3 trends were found (9% detection

rate). No trends were found for CHOCHO.

We find $NO_2$ trends are similar to those found in the default analysis. For example, trends over Tehran, Baghdad, Riydah, Kirkuk, Orumiyeh (Iran), Isfahan and Ad-Dammam are approximately the same, although at Irbil the linear trend is now 6 % $yr^{-1}$; previously it was about 9 % $yr^{-1}$.

For HCHO there were three locations which could be compared to the default case. These were: Attaif (now 3 % $yr^{-1}$ was previously 7 % $yr^{-1}$), Abu Zabi (now 2 % $yr^{-1}$ was 3% $yr^{-1}$) and Addammam (still about 3% $yr^{-1}$), but there were 6 new trend locations found with trends from 2–3 % $yr^{-1}$, consistent with HCHO trends over other urban settlements.

For $SO_2$ only Tarbiz has trend that was previously detected (61 % $yr^{-1}$), but now with a value

of 37 % $yr^{-1}$. We now find two other trends of about 15 % $yr^{-1}$ at Kermanshah (Iran) and 22 % $yr^{-1}$ at al-Mawsil (Mosul,Iraq).

In summary, for $NO_2$ and HCHO, we find that the trend detection percentages (relative to the 32 targets) increase, but generally the trends over co-existing locations found in the default analysis do not really change.

Therefore, whilst different approaches in the time series analysis may, in some cases, affect trends found over individual locations, overall there are a large number of targets (per gas species) where trends of approximately the same magnitude are consistently detected. This gives some level of confidence in the robustness of the trend analysis.

### 5.3 Comparative Studies

Another way to corroborate our results is through comparison to other similar independent studies. For example, Duncan et al. (2016) performed $NO_2$ trend analysis for the world's major cities using the official NASA product, as described in Bucsela et al. (2013), available from the NASA Goddard Earth Sciences Data Active Archive Centre (http://disc.sci.gsfc.nasa.gov). In that study, monthly mean values were based on the average of OMI data falling within $0.3° \times 0.3°$ boxes cen-

tred over the cities. We find good agreement in trend magnitudes (within cited errors) for Tehran, Baghdad, Kirkuk, Kuwait City, Beirut, Aleppo, Mecca (not classified in our analysis as a 'real' trend) and Jerusalem (not classified here as significant). There is disagreement at Mosul ($4.43 \pm 1.18$ % $yr^{-1}$, here $2.14 \pm 1.02$ % $yr^{-1}$) and Riyadh ($0.00 \pm 1.19$ % $yr^{-1}$, here $3.27 \pm 0.85$ % $yr^{-1}$), Homs ($-0.81 \pm 0.99$ % $yr^{-1}$, here $1.26 \pm 0.74$ % $yr^{-1}$ and classified as real but not significant), and Dam-

ascus ($-3.72 \pm 1.10$ % $yr^{-1}$, here $-0.65 \pm 0.92$ % $yr^{-1}$ classified as not significant or real). These





small discrepancies likely occur because of the different choices of OMI data set and different analysis methodology.

Similarly, a top-down multi-species inversion involving OMI DOMINO $NO_2$ columns by Miyazaki et al. (2016) indicated $NO_x$ emissions trends of 3.7 % $yr^{-1}$ for Tehran and 4.7 % $yr^{-1}$ for Kuwait City; which are broadly consistent with our results. However, Miyazaki et al. (2016) also found an emissions trend of $-6.0$ % $yr^{-1}$ for Dubai, whereas we find a $NO_2$ trend of $-0.56\pm1.05$ % $yr^{-1}$ (classed as real but not significant).

We have also compared our results with the global catalogue of $SO_2$ emission sources produced by Fioletov et al. (2016). Over the Middle East we were able to identify thirty corresponding targets, for which we compared the linear trends of the derived $SO_2$ emissions to our derived $SO_2$ trace gas trends. We find five locations where the $SO_2$ trends in both studies agree (within errors), and where we class the trends as real and significant. For example, there is excellent agreement at Kharg Island (here $-6.89\pm2.20\%$ $yr^{-1}$ versus $-7.84\pm2.50$ % $yr^{-1}$) and the Besat power plant in Tehran (in this study $3.54\pm3.35\%$ $yr^{-1}$, versus $3.86\pm3.61\%$ $yr^{-1}$). However, we also find good agreement at an additional 16 sites, where we class the trends as real but not significant, thus indicating the good correspondence between the two independent studies.

Lastly, both Lelieveld et al. (2015) and Duncan et al. (2016) have used independent economic and social information to attempt an interpretation of observed changes in OMI $NO_2$ and $SO_2$ over the Middle East. We have further compared our derived trends to the linear growth in the Organisation of Petroleum Exporting Countries (OPEC) oil production and demand data, and additionally, population, GDP per capita (GDP), and energy consumption per capita (EC) data, from the World Data Bank (http://data.worldbank.org). We find such comparisons, are at the very least, difficult to interpret. For example, over 2005–2014 Iran's GDP grew by 2.8 % $yr^{-1}$, inline with the $NO_2$ and HCHO median linear trends of 2–4 % $yr^{-1}$, but its oil production fell by $-2.5$ % $yr^{-1}$ conflicting with the increasing OMI trace gas trends found over refineries.

## 6 Summary and Outlook

We have performed a robust and detailed time series analysis on 10 years of OMI trace gas observations of $NO_2$, HCHO, $SO_2$, and CHOCHO, to assess changes in local air-quality for over 1000 urban, oil and energy target locations over the Middle East during the period 2005–2014.

We find the highest average pollution levels of HCHO, $SO_2$, and CHOCHO are over the major oil ports and refineries, compared to urban areas and power plants. For HCHO and CHOCHO, the average trace columns are about 15–25% higher over the ports and refineries, whereas for $SO_2$ the columns are about 60–80% higher. In contrast, $NO_2$ is found to be slightly higher over the urban areas and ports by about 5–15%. The highest average pollution levels over urban settlements are typically in Bahrain, Kuwait, Qatar, and UAE. Other notable pollutant hotspots include: (1) Kuwait





City, Tehran and Mecca; (2) the west coast of Yemen where HCHO levels are high, and (3) elevated $SO_2$ near Rasanjan and over Kharg Island. We find the observed vertical columns can exceed average levels by about 40–320%, depending on the trace gas species.

Our analysis shows that $NO_2$ linear trends over urban locations range from about $-3$ to $12 \%$ yr$^{-1}$, although only two locations showed a decrease in $NO_2$. Linear trends over oil refineries and ports are about 2–6 % yr$^{-1}$ and 2–9 % yr$^{-1}$, respectively. Trends over power plants range from 2–8 % yr$^{-1}$, with 5 of the topmost 10 trends found in Iran. For HCHO, we find urban trends are 2–7 % yr$^{-1}$. Only six trends of 2–3.5 % yr$^{-1}$ were detected over oil refineries, and only four trends of 2–4 % yr$^{-1}$ were detected over oil ports. Trends over power plants ranged from 2–7 % yr$^{-1}$. The increasing HCHO trends are mostly found along the western Gulf coast, particularly along the Saudi Arabian coast near Ad-Dammam, and also near Doha in Qatar. Very few $SO_2$ trends were detected. Over urban areas, trends ranged from about $-60$ to $120 \%$ yr$^{-1}$, with 11 of the 18 trends were detected in Iran. However, some of the high relative $SO_2$ trends can be attributed to median levels of approximately zero over the location. Over refineries, three trends were detected of about 9–15 % yr$^{-1}$, whilst the Iranian ports of Bandar-e Khomeni and Kharg Island, showed decreases of about 6% yr$^{-1}$. Apart from two locations, we find CHOCHO levels are not changing over the Middle East. Derived growth rates, are, on average, very similar to the fitted linear trends, although difference can occur for individual locations, particularly for $SO_2$. We find our derived linear trends are generally consistent with other independent OMI trend studies over this region.

Therefore, based on this analysis, we can conclude that for a large number of locations, air-quality has deteriorated over 2005–2014. Whilst effective regulatory measures have been established in some countries, it is clear that effective pollutant emission controls are required to limit health impacts on the region's population. Whether this goal is achievable is an open question, especially in countries experiencing civil unrest (e.g., Iraq, Syria, Yemen, Palestine) or increasing economic growth (e.g., Iran).

In the near future, the TROPospheric Ozone Monitoring Instrument (TROPOMI), which is expected to be launched in 2016 and has a smaller spatial observational footprint compared to OMI, should further help evaluate changes in air-quality at local scales. However, it is important that such future measurements are carefully validated and integrated with in-situ measurements and chemical transport models, so that they can be utilised properly to influence air-quality policy decision-making in Middle Eastern countries.

*Acknowledgements.* This research used the ALICE and SPECTRE High Performance Computing Facility at the University of Leicester. We acknowledge the free use of DOMINO tropospheric $NO_2$ column data from www.temis.nl, and the HCHO and $SO_2$ data from the NASA OMI portal (http://disc.sci.gsfc.nasa.gov/Aura/data-holdings/OMI).



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



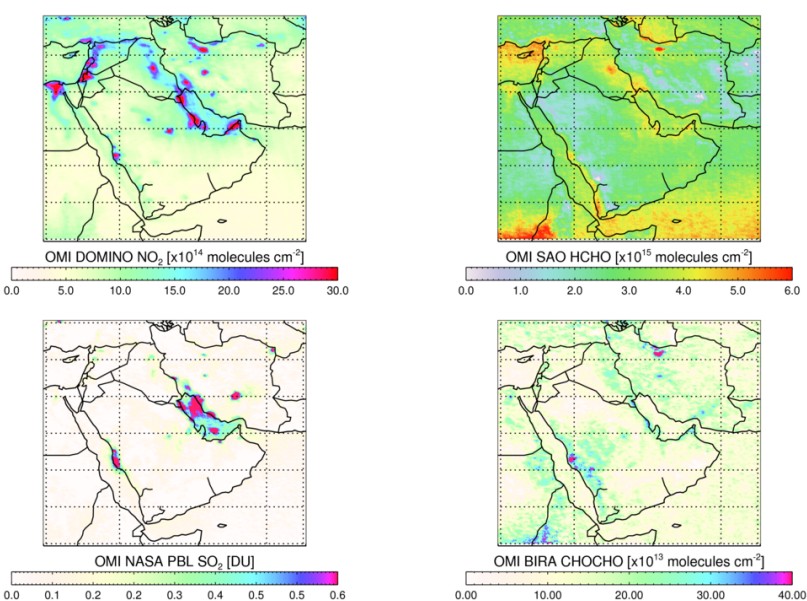

**Figure 1.** The OMI 2005 annual mean distributions of $NO_2$, HCHO, $SO_2$, and CHOCHO. The OMI data have been averaged onto a $0.05° \times 0.05°$ grid using observations with cloud fractions $< 20\%$ and and solar zenith angles $\leq 70°$. Observations affected by the row-anomaly are excluded, as described in Section 3.1, and the gridded data have been smoothed with a $0.15° \times 0.15°$ Gaussian filter of 1-$\sigma$ width (2-$\sigma$ for CHOCHO).





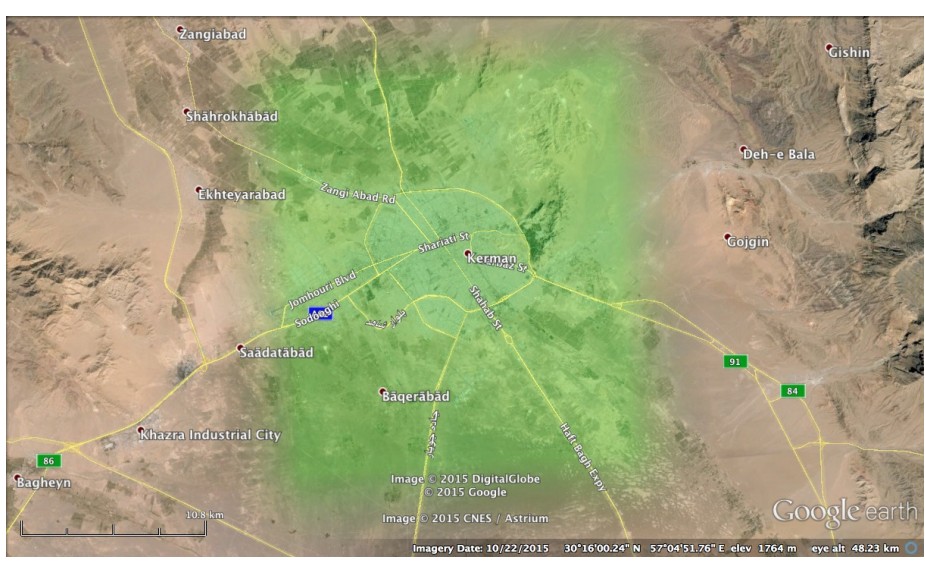

**Figure 2.** Example of GoogleEarth imagery showing the $\pm 2$ grid-cell spatial filtering mask (in light green) relative to the centre of the city of Kerman (Iran), as discussed in Section 3.2. The mask corresponds to an approximate radial distance of about 10 km.





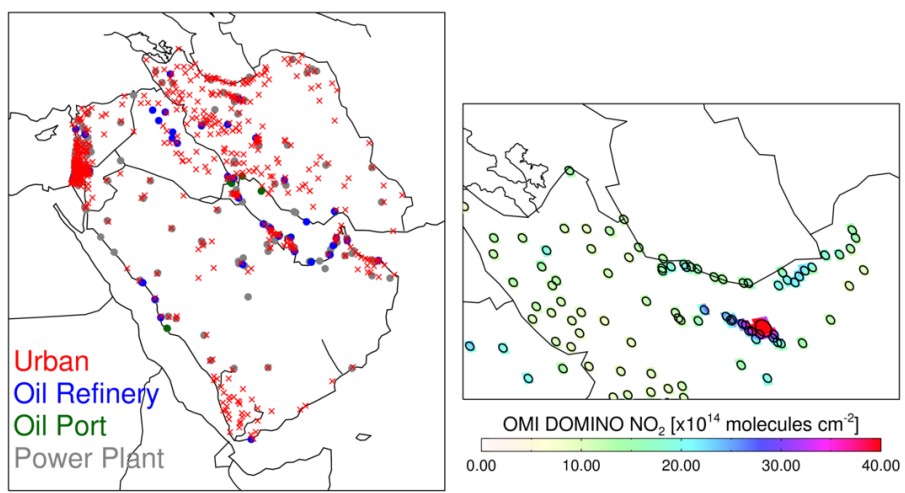

**Figure 3.** Left: the geographical distributions of the target locations showing cities/towns (in red) as specified by the GRUMP v1 settlement point data base (Balk et al., 2006; SEDAC, 2015) oil refineries (in blue) (based on Kootungal, 2010), and oil ports and power plants based on the Global Energy Observatory (GEO) online resource (http://globalenergyobservatory.org). Right: OMI 2005 $NO_2$ annual mean of over northern Iran, with a spatial filtering mask applied to extract observations over urban centres (see Section 3.2). The OMI data have been averaged onto a $0.05° \times 0.05°$ grid using observations with cloud fractions $< 20\%$ and and solar zenith angles $\leq 70°$. Observations affected by the row-anomaly are excluded, as described in Section 3.1, and have been smoothed with a $0.15° \times 0.15°$ Gaussian filter of 1-$\sigma$ width.





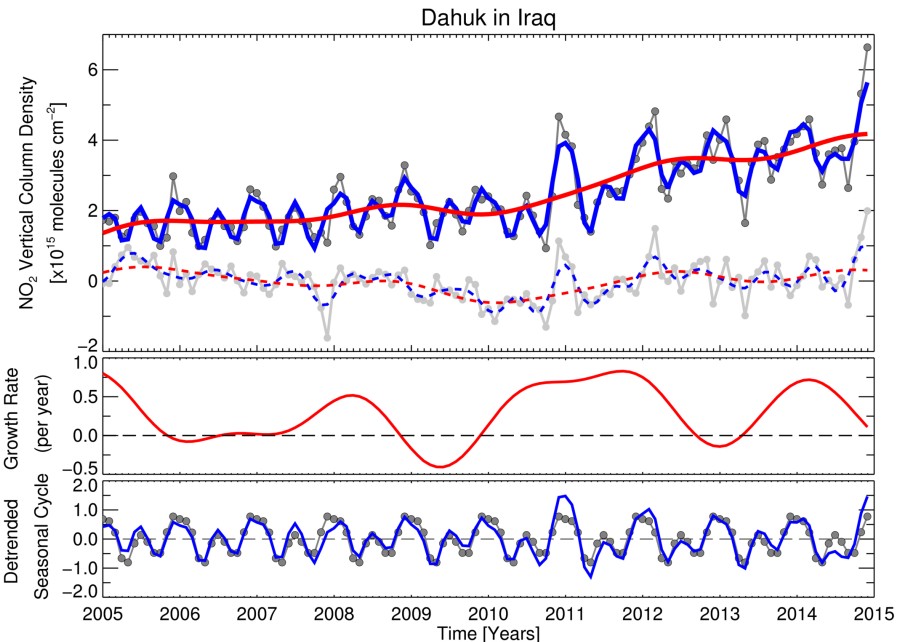

**Figure 4.** An example of a time series fit to observed $NO_2$ data over Dahuk, Iraq, as outlined in Section 3.3. Top panel: the monthly OMI $NO_2$ vertical columns are indicated by dark grey filled circles, whilst the light grey filled circles represent the fitting residual, which has been smoothed with a short-term 200-day filter (dashed blue line) and long-term 667-day filter (red dashed line). The solid red line is the long-term trend $F_T(t)$, given by the linear component of the fitted function $F(t)$ (equation 1) plus the residual filtered using the long-term filter. The solid blue line is the smoothed fitted curve $F_S(t)$ given by $F(t)$ plus the residual filtered using the short-term filter. Middle panel: The $NO_2$ vertical column growth rate in $\times 10^{15}$ molecules cm$^{-2}$ yr$^{-1}$, which is the derivative of the long-term trend $F_T(t)$ shown in the top-panel. Bottom panel: the de-trended seasonal cycle $F_C(t)$ which is the difference between the long-term trend and the smoothed function fit (i.e. $F_S(t) - F_L(t)$). This represents the annual seasonal oscillation with any long-term trend removed. The dark grey filled circles are the fitted harmonic components of $F(t)$.



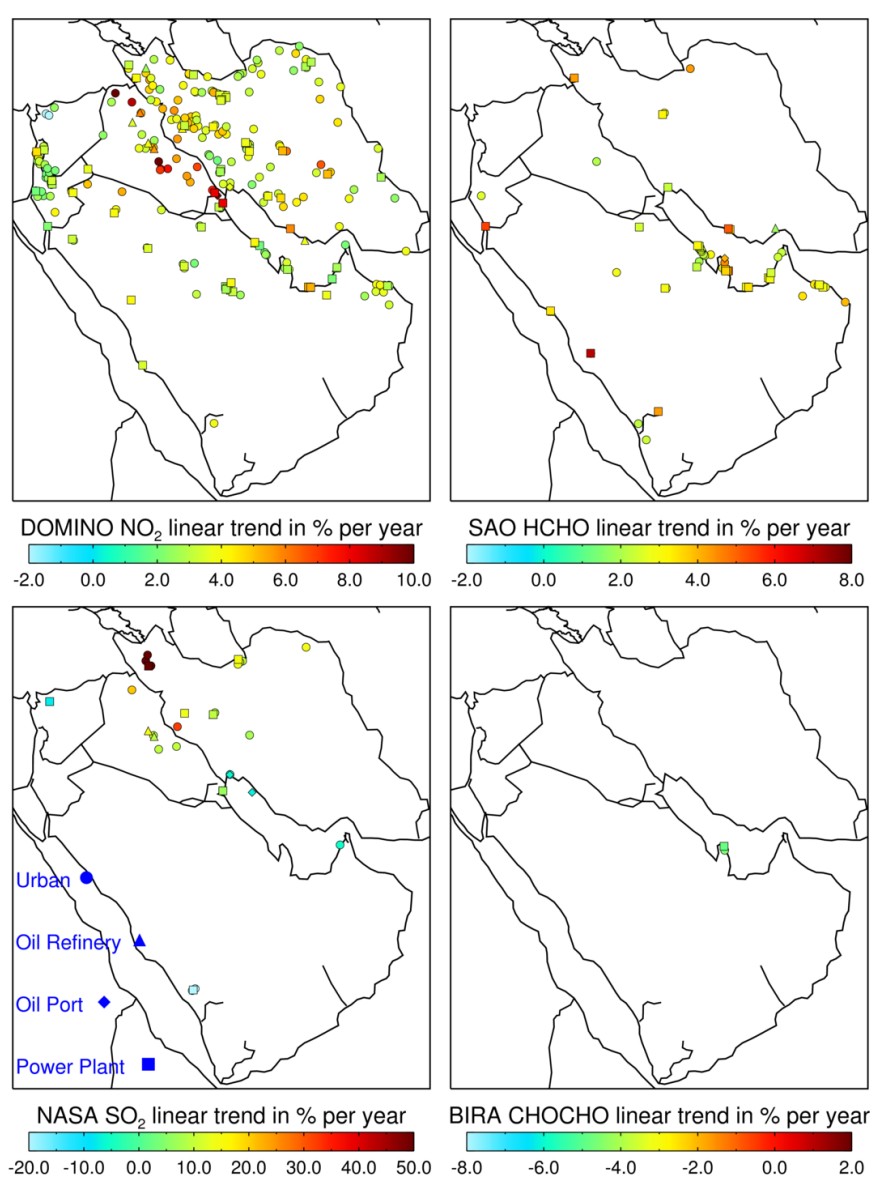

**Figure 5.** Geographical distribution of locations with statistically significant linear trends in $NO_2$ (top left), HCHO (top right), $SO_2$ (bottom left), and CHOCHO (bottom right) expressed in percent per year, relative to each location's 2005–2014 median vertical column.





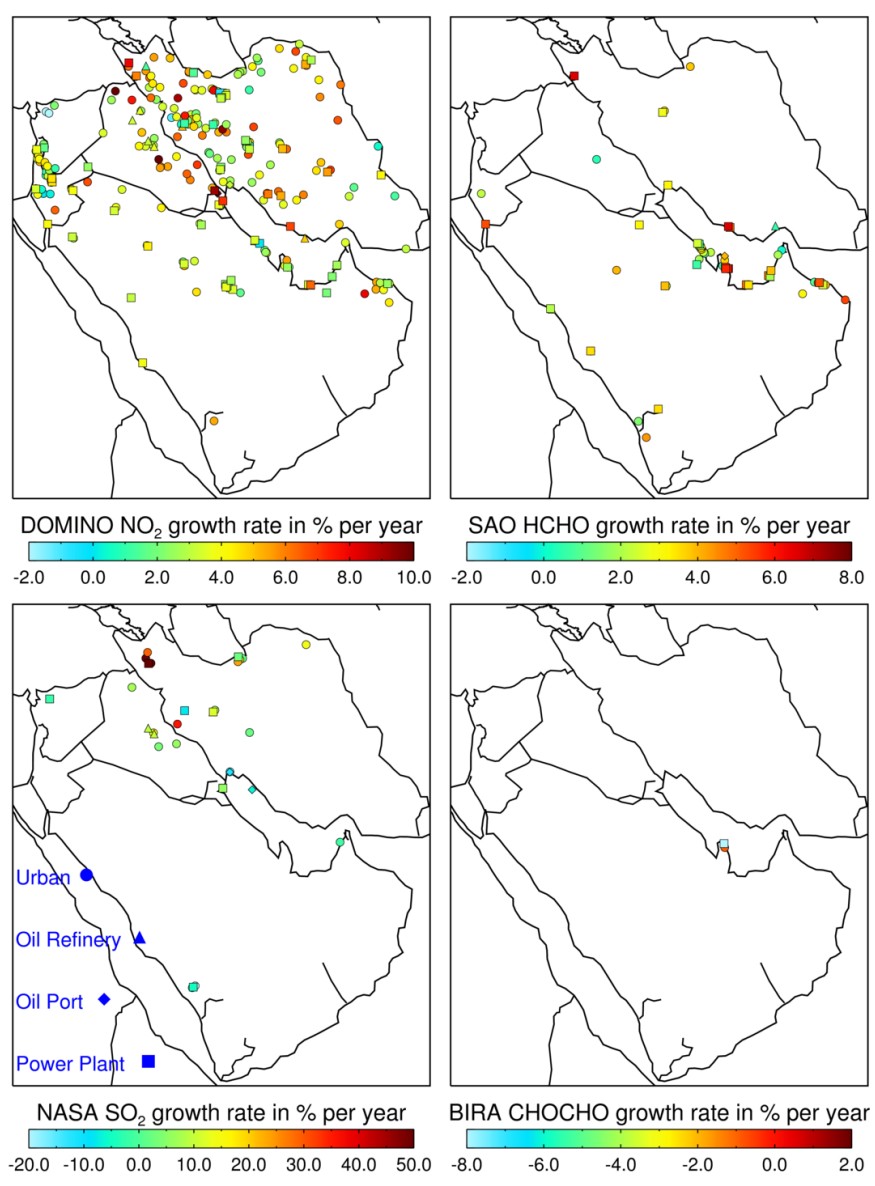

**Figure 6.** Geographical distribution of locations with statistically significant linear trends in $NO_2$ (top left), HCHO (top right), $SO_2$ (bottom left), and CHOCHO (bottom right) but here showing the growth rate expressed in percent per year, relative to each location's 2005–2014 median vertical column.





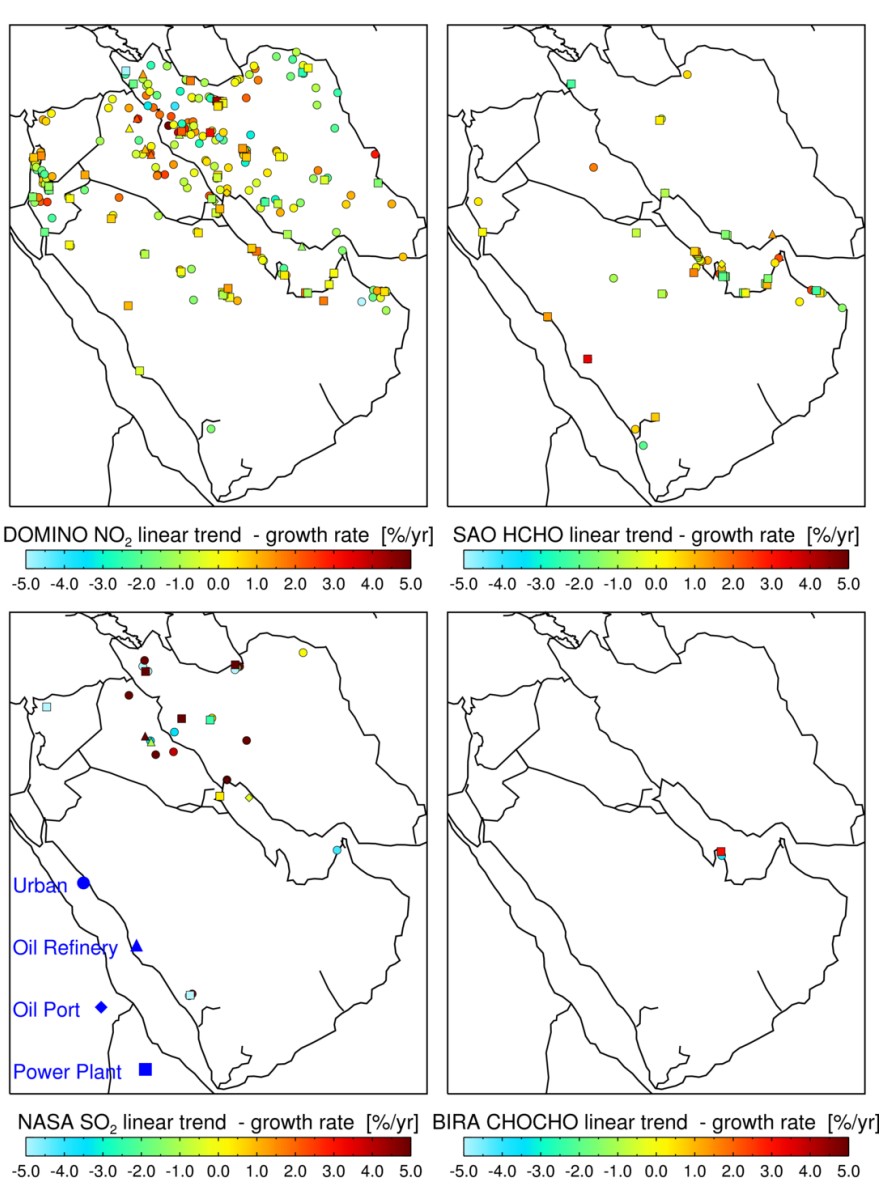

**Figure 7.** Geographical distribution of locations with statistically significant linear trends in $NO_2$ (top left), HCHO (top right), $SO_2$ (bottom left), and CHOCHO (bottom right), but here showing the linear trend (in percent per year) minus the growth rate (in percent per year).





**Table 1.** Statistical summary of the DOMINO $NO_2$ vertical column density (VCD) analysis conducted at an approximate 10 km radius over the target locations. Statistical values are shown for those locations which have a real VCD trend, i.e. those locations without a corresponding trend in either the air mass factor (AMF), cloud fraction (CFR), cloud-top pressure (CTP), or number of samples (SAM). Values in parentheses correspond to statistics for all locations of a given category. VCD values are given in $10^{14}$ molecules $cm^{-2}$ and IQR is the inter-quartile range.

|  | Urban | Refinery | Oil Ports | Power Plants |
|---|---|---|---|---|
| # locations | 818 | 41 | 18 | 155 |
| **# locations with trends** |  |  |  |  |
| VCD | 318 | 32 | 13 | 112 |
| VCD and AMF | 71 | 7 | 2 | 19 |
| VCD and CFR | 24 | 6 | 3 | 19 |
| VCD and CTP | 14 | 5 | 2 | 9 |
| VCD and SAM | 35 | 5 | 2 | 18 |
| VCD only | 198 | 17 | 6 | 57 |
| **Observed VCD** |  |  |  |  |
| Median | 18.27 (28.31) | 26.74 (27.10) | 33.00 (28.19) | 27.22 (24.32) |
| IQR | 15.00 (18.29) | 15.78 (15.78) | 11.45 (12.18) | 14.20 (14.16) |
| Maximum | 136.65 (136.65) | 87.90 (87.90) | 64.92 (100.72) | 189.85 (189.85) |
| Minimum | -4.87 (-10.61) | -6.63 (-6.63) | 10.55 (5.94) | -6.63 (-6.63) |
| **Seasonal Amplitude** |  |  |  |  |
| Median | 8.82 (12.38) | 11.86 (11.86) | 11.54 (11.83) | 11.02 (10.52) |
| IQR | 5.30 (8.01) | 9.78 (8.77) | 6.93 (5.28) | 6.03 (6.60) |
| Maximum | 38.88 (38.88) | 33.05 (33.05) | 18.50 (21.30) | 46.32 (66.32) |
| Minimum | 3.07 (1.49) | 4.62 (3.28) | 7.05 (5.76) | 4.34 (1.68) |
| **Linear Trend** |  |  |  |  |
| Median | 0.58 (0.26) | 0.84 (0.81) | 1.05 (0.97) | 0.89 (0.65) |
| IQR | 0.48 (0.49) | 0.69 (0.66) | 2.16 (1.26) | 0.45 (0.68) |
| Maximum | 2.77 (3.14) | 2.65 (2.65) | 3.12 (3.35) | 2.65 (3.34) |
| Minimum | -0.75 (-0.75) | 0.57 (-0.23) | 0.74 (-0.01) | 0.25 (-0.72) |
| **Growth Rate** |  |  |  |  |
| Median | 0.64 (0.42) | 0.83 (0.70) | 0.95 (0.89) | 0.82 (0.62) |
| IQR | 0.56 (0.52) | 0.71 (0.95) | 2.59 (1.68) | 0.64 (0.83) |
| Maximum | 2.82 (4.89) | 1.74 (2.55) | 3.44 (4.31) | 2.95 (3.91) |
| Minimum | -0.91 (-0.98) | 0.19 (-0.57) | 0.51 (0.14) | -0.33 (-0.84) |
| **Median Errors** |  |  |  |  |
| Linear trend | 0.19 (0.27) | 0.27 (0.27) | 0.29 (0.28) | 0.27 (0.25) |
| Trend Curve | 0.80 (1.13) | 1.13 (1.13) | 1.20 (1.16) | 1.14 (0.99) |
| Smoothed Curve | 1.13 (1.68) | 1.38 (1.63) | 1.84 (1.78) | 1.70 (1.53) |
| Growth Curve | 1.14 (1.60) | 1.59 (1.59) | 1.70 (1.64) | 1.61 (1.40) |
| Seasonal Curve | 1.39 (2.02) | 1.74 (1.95) | 2.20 (2.13) | 2.03 (1.84) |



**Table 2.** Statistical summary of the SAO HCHO vertical column density (VCD) analysis conducted at an approximate 10 km radius over the target locations. Statistical values are shown for those locations which have a real VCD trend, i.e. those locations without a corresponding trend in either the air mass factor (AMF), cloud fraction (CFR), cloud-top pressure (CTP), or number of samples (SAM). Values in parentheses correspond to statistics for all locations of a given category. VCD values are given in $10^{15}$ molecules cm$^{-2}$ and IQR is the inter-quartile range.

| | Urban | Refinery | Oil Ports | Power Plants |
|---|---|---|---|---|
| # locations | 818 | 41 | 18 | 155 |
| **# locations with trends** | | | | |
| VCD | 63 | 13 | 6 | 40 |
| VCD and AMF | 9 | 1 | 0 | 6 |
| VCD and CFR | 8 | 4 | 0 | 9 |
| VCD and CTP | 0 | 0 | 0 | 0 |
| VCD and SAM | 16 | 4 | 2 | 6 |
| VCD only | 34 | 6 | 4 | 26 |
| **Observed VCD** | | | | |
| Median | 4.36 (3.73) | 3.95 (4.12) | 4.51 (4.32) | 4.13 (3.70) |
| IQR | 1.03 (0.53) | 0.67 (0.78) | 0.80 (0.51) | 0.88 (0.95) |
| Maximum | 8.99 (10.49) | 7.95 (9.10) | 8.58 (8.91) | 8.89 (9.22) |
| Minimum | -0.80 (-3.11) | 0.84 (-1.53) | 1.00 (0.15) | -0.80 (-1.53) |
| **Seasonal Amplitude** | | | | |
| Median | 3.19 (3.13) | 2.85 (3.27) | 3.45 (3.43) | 3.07 (3.02) |
| IQR | 0.85 (1.09) | 0.65 (1.15) | 0.70 (1.31) | 0.69 (0.95) |
| Maximum | 4.15 (5.65) | 3.27 (4.54) | 4.08 (4.40) | 4.54 (4.89) |
| Minimum | 1.51 (1.37) | 2.46 (1.55) | 3.17 (2.33) | 1.83 (1.20) |
| **Linear Trend** | | | | |
| Median | 0.11 (0.02) | 0.12 (0.07) | 0.14 (0.08) | 0.13 (0.04) |
| IQR | 0.04 (0.06) | 0.05 (0.06) | 0.04 (0.06) | 0.06 (0.09) |
| Maximum | 0.22 (0.22) | 0.15 (0.18) | 0.16 (0.16) | 0.22 (0.22) |
| Minimum | 0.09 (-0.12) | 0.07 (-0.02) | 0.11 (-0.01) | 0.08 (-0.11) |
| **Growth Rate** | | | | |
| Median | 0.13 (0.01) | 0.08 (0.08) | 0.15 (0.10) | 0.16 (0.05) |
| IQR | 0.09 (0.10) | 0.04 (0.10) | 0.06 (0.14) | 0.11 (0.13) |
| Maximum | 0.33 (0.33) | 0.15 (0.26) | 0.16 (0.18) | 0.32 (0.32) |
| Minimum | -0.00 (-0.50) | 0.02 (-0.06) | 0.08 (-0.04) | 0.02 (-0.15) |
| **Median Errors** | | | | |
| Linear trend | 0.05 (0.05) | 0.04 (0.05) | 0.05 (0.05) | 0.05 (0.05) |
| Trend Curve | 0.19 (0.23) | 0.19 (0.20) | 0.20 (0.21) | 0.20 (0.21) |
| Smoothed Curve | 0.30 (0.35) | 0.30 (0.32) | 0.31 (0.32) | 0.31 (0.32) |
| Growth Curve | 0.27 (0.32) | 0.27 (0.29) | 0.28 (0.29) | 0.28 (0.29) |
| Seasonal Curve | 0.36 (0.41) | 0.36 (0.38) | 0.36 (0.38) | 0.37 (0.38) |





**Table 3.** Statistical summary of the NASA $SO_2$ vertical column density (VCD) analysis conducted at an approximate 10 km radius over the target locations. Statistical values are shown for those locations which have a real VCD trend, i.e. those locations without a corresponding trend in either the cloud fraction (CFR), cloud-top pressure (CTP), or number of samples (SAM). Values in parentheses correspond to statistics for all locations of a given category. VCD values are given in DU, and IQR is the inter-quartile range.

|  | Urban | Refinery | Oil Ports | Power Plants |
|---|---|---|---|---|
| # locations | 818 | 41 | 18 | 155 |
| **# locations with trends** |  |  |  |  |
| VCD | 24 | 3 | 3 | 14 |
| VCD and AMF | 0 | 0 | 0 | 0 |
| VCD and CFR | 0 | 0 | 0 | 0 |
| VCD and CTP | 0 | 0 | 0 | 0 |
| VCD and SAM | 6 | 0 | 1 | 5 |
| VCD only | 18 | 3 | 2 | 9 |
| **Observed VCD** |  |  |  |  |
| Median | 0.17 (0.11) | 0.30 (0.28) | 0.63 (0.28) | 0.29 (0.16) |
| IQR | 0.23 (0.09) | 0.33 (0.31) | 0.11 (0.21) | 0.27 (0.24) |
| Maximum | 1.22 (1.79) | 1.05 (1.85) | 1.98 (1.98) | 1.17 (2.22) |
| Minimum | -0.61 (-1.36) | -0.22 (-0.64) | -0.02 (-0.40) | -0.46 (-0.46) |
| **Seasonal Amplitude** |  |  |  |  |
| Median | 0.33 (0.21) | 0.36 (0.44) | 0.55 (0.39) | 0.37 (0.28) |
| IQR | 0.21 (0.09) | 0.13 (0.26) | 0.11 (0.20) | 0.19 (0.28) |
| Maximum | 0.69 (1.15) | 0.47 (1.13) | 0.77 (0.86) | 0.51 (1.14) |
| Minimum | 0.17 (0.09) | 0.34 (0.16) | 0.33 (0.20) | 0.18 (0.12) |
| **Linear Trend** |  |  |  |  |
| Median | 0.021 (-0.002) | 0.028 (0.001) | -0.041 (-0.002) | 0.026 (-0.001) |
| IQR | 0.023 (0.008) | 0.022 (0.023) | 0.009 (0.025) | 0.046 (0.011) |
| Maximum | 0.044 (0.044) | 0.042 (0.042) | -0.024 (0.028) | 0.028 (0.031) |
| Minimum | -0.024 (-0.041) | 0.019 (-0.016) | -0.059 (-0.059) | -0.023 (-0.023) |
| **Growth Rate** |  |  |  |  |
| Median | 0.015 (-0.002) | 0.036 (0.003) | -0.052 (-0.002) | 0.005 (-0.003) |
| IQR | 0.032 (0.011) | 0.034 (0.027) | 0.002 (0.034) | 0.032 (0.017) |
| Maximum | 0.048 (0.051) | 0.046 (0.046) | -0.049 (0.047) | 0.036 (0.057) |
| Minimum | -0.056 (-0.056) | 0.012 (-0.045) | -0.055 (-0.055) | -0.012 (-0.039) |
| **Median Errors** |  |  |  |  |
| Linear trend | 0.009 (0.007) | 0.010 (0.010) | 0.014 (0.010) | 0.010 (0.008) |
| Trend Curve | 0.039 (0.030) | 0.042 (0.041) | 0.062 (0.044) | 0.039 (0.035) |
| Smoothed Curve | 0.058 (0.046) | 0.066 (0.061) | 0.091 (0.068) | 0.064 (0.053) |
| Growth Curve | 0.056 (0.042) | 0.060 (0.058) | 0.087 (0.063) | 0.056 (0.050) |
| Seasonal Curve | 0.070 (0.055) | 0.079 (0.072) | 0.110 (0.081) | 0.075 (0.064) |





**Table 4.** Statistical summary of the BIRA CHOCHO vertical column density (VCD) analysis conducted at an approximate 10 km radius over the target locations. Statistical values are shown for those locations which have a real VCD trend, i.e. those locations without a corresponding trend in either the air mass factor (AMF), cloud fraction (CFR), cloud-top pressure (CTP), or number of samples (SAM). Values in parentheses correspond to statistics for all locations of a given category. VCD values are given in $10^{13}$ molecules cm$^{-2}$ and IQR is the inter-quartile range.

| | Urban | Refinery | Oil Ports | Power Plants |
|---|---|---|---|---|
| # locations | 818 | 41 | 18 | 155 |
| **# locations with trends** | | | | |
| VCD | 8 | 0 | 0 | 4 |
| VCD and AMF | 5 | 0 | 0 | 3 |
| VCD and CFR | 1 | 0 | 0 | 1 |
| VCD and CTP | 0 | 0 | 0 | 0 |
| VCD and SAM | 1 | 0 | 0 | 0 |
| VCD only | 1 | 0 | 0 | 1 |
| **Observed VCD** | | | | |
| Median | 18.22 (16.18) | NaN (19.95) | NaN (20.84) | 18.26 (17.88) |
| IQR | NaN (4.58) | NaN (9.61) | NaN (8.72) | NaN (8.60) |
| Maximum | 33.76 (112.10) | NaN (63.18) | NaN (57.19) | 35.50 (80.37) |
| Minimum | 0.18 (-66.32) | NaN (-23.96) | NaN (-21.89) | 2.63 (-44.75) |
| **Seasonal Amplitude** | | | | |
| Median | 9.50 (15.03) | NaN (16.48) | NaN (14.38) | 11.77 (16.32) |
| IQR | NaN (6.54) | NaN (4.78) | NaN (4.80) | NaN (5.74) |
| Maximum | 9.50 (44.53) | NaN (28.34) | NaN (21.10) | 11.77 (32.14) |
| Minimum | 9.50 (8.89) | NaN (9.56) | NaN (9.93) | 11.77 (9.12) |
| **Linear Trend** | | | | |
| Median | -0.83 (-0.22) | NaN (-0.05) | NaN (-0.01) | -0.88 (-0.10) |
| IQR | NaN (0.66) | NaN (0.49) | NaN (0.53) | NaN (0.57) |
| Maximum | -0.83 (3.02) | NaN (0.40) | NaN (0.30) | -0.88 (0.87) |
| Minimum | -0.83 (-1.38) | NaN (-1.03) | NaN (-0.77) | -0.88 (-1.29) |
| **Growth Rate** | | | | |
| Median | -0.16 (-0.38) | NaN (-0.12) | NaN (-0.13) | -1.46 (-0.22) |
| IQR | NaN (1.06) | NaN (0.98) | NaN (0.83) | NaN (1.10) |
| Maximum | -0.16 (3.85) | NaN (1.26) | NaN (0.83) | -1.46 (2.05) |
| Minimum | -0.16 (-2.96) | NaN (-1.70) | NaN (-1.52) | -1.46 (-1.95) |
| **Median Errors** | | | | |
| Linear trend | 0.42 (0.58) | NaN (0.53) | NaN (0.50) | 0.44 (0.55) |
| Trend Curve | 1.79 (2.49) | NaN (2.28) | NaN (2.11) | 1.83 (2.41) |
| Smoothed Curve | 2.84 (3.95) | NaN (3.59) | NaN (3.38) | 2.79 (3.66) |
| Growth Curve | 2.53 (3.52) | NaN (3.22) | NaN (2.98) | 2.58 (3.40) |
| Seasonal Curve | 3.35 (4.68) | NaN (4.23) | NaN (3.99) | 3.34 (4.38) |