# Peer review of "OMI air-quality monitoring over the Middle East"

_Atmospheric Chemistry and Physics, 2016_

## Referee Comment (RC1) · Anonymous Referee #2 · 10 Nov 2016

The paper contains a very thorough analysis of some air quality gases in the Middle East. It is scientifically speaking not very new, but it is a very useful overview of the air quality in the region with many details.

Comments

Line 160: Scenes with effective cloud fraction > 20 % are rejected for CHOCHO, in line 177-178: a filter for cloud radiance fraction > 0.3 is mentioned for SO2, while in line188 it is mentioned that scenes with a higher than 20% fractional cloud cover are filtered for all data. Why are you not using a single cloud filter setting for all data.

Line 180: Several times in the period of this research volcanoes in North-East Africa have been erupting with as result volcanic plumes over the Middle East region. Large parts of the plume have values of less than 5DU. For example in June 2011 many days

show remnant SO2 values caused by the eruption of the Nabro volcano. I would expect this affects the trend considerably and it might be better to remove this period from the time series of SO2.

Line 200-201: Does this mean that for the gases HCHO and NO2 the row anomaly mask in 2013 is less strict than in some earlier years? By using different criteria over the years for these gases your trend is still affected by sampling I would think. Please clarify this section.

Line 224: I think this Figure 2 does not add much to the paper and in my view can be removed.

Line 579: Also for NO2 a cloud fraction of less than 20% is advised. However, I doubt these advices were given with this particular test in mind. I suggest to apply this test also to CHOCHO and SO2.

Line 595-599: I thought the standard analysis was already done for unaffected rows. How are the unaffected rows defined in this particular test ?

Line 622: It is not only the number of observations but also the type. Because you are using a different selection of rows you have a selection of other pixel sizes. The change of pixel sixe alone will already affect the derived trends. This should be added to the discussion.

Section 5.3: In this section it might be interesting to include the study of Schneider et al. (2015), who did a trend analysis on NO2 in large urban agglomerations for the period 2002-2012, based on SCIAMACHY.

Figure 3, left-side: In this Figure I have difficulties distinguishing between oil refineries, oil ports or power plants. Other symbols or colors can improve the Figure. Please also add the symbols to the legend.

Table 1-4: In my opinion large part of this table can be moved to the supplementary material.

In general: the significance in which most values are given is much too high compared to their errors. A digit less is often possible and makes the text better readable.

---

## Referee Comment (RC2) · Anonymous Referee #1 · 6 Dec 2016

[referee-annotated manuscript omitted]

---

## Author Comment (AC1) · 30 Jan 2017

**Response to Reviewers**

We thank both reviewers for their supportive and insightful comments. We have addressed all issues and have adjusted the manuscript accordingly where necessary. Our responses are provided in blue italic. We have also attached the revised manuscript with the tracked changes in blue.

**Reviewer 1**

The authors of OMI air-quality monitoring over the Middle East have made a great effort to take on board most of my original comments, especially with regard to the length of the supplementary material. The presentation of their work has increased overall, which will lead to increased visibility to the community. No problem – thanks for the helpful suggestions!

Section 2: Please add references of validation/cross-comparison works of the satellite products where appropriate. We have added key validation references for HCHO and NO2, but note there are no validation references for the new BIRA OMI CHOCHO product. Given the manuscript length, we don't think adding a paragraph describing applications of the DOMINO NO2 product is critical to the paper (especially as a reader can refer to the already cited references and thus additional references therein).

Line 177: Did you use similar cut-off criteria for the other species? if so, please add in the relevant paragraphs. A cloud radiance fraction of 30% corresponds to an effective cloud fraction of about 15-20% [Stammes et al. 2008, Krotkov et al., 2016], making it broadly consistent with cloud filters applied to the other species. We have added this detail to the text.

Line 181: Is this for each individual measurement? the value seems rather small. Please expand this phrase accordingly.

The SO2 data are in this study are assigned a default uncertainty of 0.5 DU before gridding (as individual PBL SO2 measurements in the product do not have an associated error). This value is taken from the NASA SO2 ReadMe file, which is based on the analysis of the root mean square (RMS) and standard deviation values for instantaneous field of view (IFOV) observations in different latitudinal bands. In the tropics the error is about 0.5 DU but can be up to 0.7-0.9 at higher latitudes where there is strong ozone interference. We have added this information to the text.

Line 210: Please explain why in the HCHO plot there appear to be high values in the bottom of the domain, also around the eastern Mediterranean and the island of Cyprus?

In the bottom of the domain, high HCHO columns occur due to the oxidation of biogenic volatile organic compounds, emitted predominately by vegetation and from fires within the African tropics [see e.g., Marias et al., 2013]. Over the eastern Mediterranean Sea, the high HCHO columns are a known retrieval artifact owing to the presence of Saharan dust compromising the AMF calculations, and also a potentially an unknown HCHO source, as discussed in the study of Sabolis et al. [2011]. Hence these features are well-documented within the literature.

Line 242: Please discuss the differences between these medians and the ones noted above, also giving the 1sigma. Are these medians only of the locations or the entire domain?

In the previous section 3.1, the median uncertainties correspond to the errors of the gridded data (i.e. at 0.05x0.05 degree resolution). As explained in section 3.2, for each location we construct a time series of the monthly averaged vertical columns, based on the weighted averaged of the gridded cells around the target. Thus, every time series data point has a corresponding uncertainty. We simply calculate the median of these vertical column time series data point uncertainties, for each species over all locations. We have adjusted the text to "...median uncertainties of the trace gas vertical column time series **data points** reduce..." to hopefully make this clearer. We have also now added the 1-sigma error to each of these median values.

Line 248: Why didn't you use the IQR as filter? since you have used it above as statistical tool. Did you try different cut-offs and found that the 3 median STDs is the best? also, did you filter with this method, when you calculated the percentages shown in line 244? please explain.

We use the median absolute deviation as we have previously found it a more robust statistical filter for removing outliers in satellite trace gas columns (particularly HCHO), than other types of methods (e.g., using the IQR or standard-deviation). Note there is a typo in the original manuscript - the cut-off is actually 2.5 - as recommended in the cited reference: Leys et al. [2013]. Apologies. Various filter cut-off values were tested. Lower values (<2.5) removed too many data points to perform a meaningful analysis, whilst higher values (>2.5) did not remove all outlying peaks sufficiently. We also ask the reviewer to note that the trends are not significantly affected by this

filtering processes (see e.g., Test 5 in Section 5.2). The uncertainties in section 3.2 (line 244) are computed prior to filtering.

Line 250: Why not the AMFs? isn't the main error source for all the DOAS-type analysis techniques, the AMF? Agreed, the AMF is one of the main error sources in tropospheric trace gas retrievals. However, quite simply there are no outlying spikes in the AMFs to filter out as their variation is relatively smooth in comparison, hence it is not necessary step.

Line 251: 20% continuous months or random, scattered in the time series? *Randomly scattered through the time series. We have added this detail to the text.*

Line 252: 6% in a ten year long monthly mean time series, means that 7 months out of 120 months are missing in total. I find that quite hard to believe, especially for SO2 and CHOCHO. Unless your 3 median filter is too lax, or you actually mean something else and not 6% in total. For e.g. the SO2 NASA product comes with the specification that the winter months not be used, I am assuming that you used them anyway. And even though the uncertainties are strong, there were still no more than 7 months out of 120 missing? Please explain and expand on this issue.

The 6% is simply the largest median number of points missing per time series out of all four species and four target categories (here quoted for NO2). However, you are quite right in some cases the number of points missing can be notably higher, e.g., between 0-25%. We now make this clearer in the text.

Line 307: This phrase is rather difficult to follow, too dense maybe. How about re-wording it in simpler terms? *Adjusted.*

Line 312: Since some, if not all, of the main conclusions/discussion in this work is based on the trend and the growth rate, Fg, it is important to show that using 667 days compared to 700 days provides similar results. What exactly does "days" mean in this context, since your time series are in monthly means?

In our initial tests we tried different combinations of the cited long-term filter values (500, 667, and 720 days) for the DOMINO NO2 and SAO HCHO products over urban areas (818 targets) to assess their impact on the growth (note the filters do not affect the fitted linear trend). We found the differences in the growth rates small using the 500 day and 720 day filters, compared with the default 667-day filter. For NO2, the differences are less than 1%, and for HCHO they are 1-2%. We have added this information to the text. The filters are given in days but are converted to months in the analysis code.

Line 315: For such a small town, of 65000 people, how do you explain such an increase in NO2? surely not traffic. Are there power plants around?

Although accurate statistics are difficult to find, it is believed that during the last decade or so, Dahuk itself experienced a sizeable urban growth [Mustafa et al. 2012], and now has a population of about 280,000 [MOP-KRG, 2012], while the region itself has had to accommodate an influx (>100,000) of Syrian refugees [UNOCHA]. Both factors likely contribute to the growth in NO2 levels. We now mention this in the text and added appropriate references.

Line 318: Is this value considered good enough? you note above that anything higher than 2 is statistically sound, however is 8 good? or is a value of 800 expected in robust cases?

Published values of the trend ratio are typically 2-6 – see e.g., van der A et al. [2006] or De Smedt et al. [2010]. Hence we consider a value of 8 as a strong trend. Values of 800 would imply negligible noise.

Line 322: I might understand why you repeated the trend analysis for the cloud parameters, since, there might have been a trend in cloudiness indeed over the decade. Why did you perform the trend analysis on the AMF, what is the physical reason? is there a parameter that enters the AMF calculation that is expected to have a possible trend? I would be more interested in seeing whether the radiances themselves have a trend, but that is beyond the scope of this paper and something for the PIs who create the datasets to worry about.

As a precaution, we felt it was necessary to identify AMF trends to eliminate them as a cause of vertical column trends. In theory, unless there is an underlying trend in the a priori trace gas profiles and surface reflectance (typically these are fixed to a single year), or cloud parameters, we shouldn't expect any AMF trends. However, AMF trends do occur, mostly caused by the cloud parameters and/or sampling effects.

Line 336: Is this the median for the decade? for the locations only? please clarify for all the rest of the section as well. What is the STD associated with this value? aren't the HCHO and SO2 values within their individual limits of statistical significance?

Yes, these are the averages for the decade (2005-2014) determined using all 1032 target locations (i.e. urban + oil ports + refineries + power plants), and thus are statistically significant (at the 95% confidence level). We have made this point clearer in the text and added the 1-sigma errors.

Line 342: Why do you suppose this is? what is the physics behind HCHO being higher in ports? To clarify we have only examined levels of the pollutants over **oil ports** – we have corrected this throughout text. The reason why HCHO is higher over the oil ports is unclear.

Line 392: This is the average for the decade? it is extremely low, and I am guessing well outside any statistical significance. Please discuss this issue.

As discussed above, this is the average SO2 for the decade (2005-2014), and is statistically significant (at the 95% confidence level). The SO2 values for we find for the Gulf region are consistent with studies of Krotkov et al. [2016], Fioletov et al. [2016], and McLinden et al. [2016].

Line 422: I fail to understand the meaning of this. In line 392 you quote that "the highest level of 0.73 D.U. was...". In general, I found this sub-section rather tiresome and repetitive of information given already above. You can merge the two paragraphs, if you wish, so that for e.g. when you discuss median NO2 levels, you can also give the highest NO2 level.

In line 392, the 0.73 DU is the highest **median** vertical column level. In line 422, we refer to the highest **maximum** SO2 vertical columns. We have corrected the text in Section 4.2 and line 392 to avoid any misunderstanding. We prefer to keep Sections 4.1 and the 4.2 as distinct components to make it obvious that the former discusses average levels, whilst the latter only discusses maximum values.

Line 428: You mean peak-to-peak amplitude? and have you averaged all locations? only those with statistically significant seasonality? in general, this type of information is missing from the text. I understand the authors have this knowledge, but it would be good to have it in the text so that other colleagues can follow this work and compare like with like.

Apologies for not including this information at the start of this section – although we ask the reviewer to note we explicitly state how the seasonal amplitudes were calculated in Section 3.3; see line 293 of the original manuscript. Hence we have now added at the start of Section 4.3 the following: "To assess the seasonal variability over each target we determined the average peak-to-peak difference of its corresponding time series, and used these values to compute the median seasonal amplitude over all locations, within each target category." – hopefully this makes it clearer to the reader.

Line 432: So, what does this mean? that the highest amplitude is over the power plant in Iran and the lowest over Ataq? physics-wise? What is this seasonality in HCHO due to?

Yes, this is correct. We have adjusted the sentence to make it clearer. The seasonality of HCHO and CHOCHO are due to seasonal variations of biogenic VOCs, specifically isoprene whose emissions peak in July-August time and methanol. We have added this to the text and included the reference:

• Müller, J.-F., Stavrakou, T., Smedt, I. D., and Roozendael, M. V.: VOC emissions in the Middle East from bottom-up inventories & as seen by OMI, GlobEmission User Consultation Meeting, Doha, Qatar, 24-25 November, 2015.

Line 441: How is this possible, since you used FFT to de-noise and de-trend the data? I think you should reconsider this phrase and what exactly you mean.

We only meant that the SO2 and CHOCHO retrievals are noisier than their NO2 and HCHO counterparts, and thus their time series even after filtering and smoothing are noisier too. We have adjusted the text accordingly.

Line 445: Not sure I follow you here. Are there 274 towns, power plants, oil ports, and such like in Palestine? or do you mean, 274 pixels that comprise Palestine? please rephrase accordingly.

We do not understand the confusion. It is clearly stated in the manuscript that "...if we disregard Palestine, where only 1 out of its 274 urban targets had a real trend..." – i.e. we are not discussing gridded pixels. Nevertheless, we have changed the text to: "However, in Palestine only 1 out of a potential 274 urban targets had a real trend. Neglecting the Palestine results in this instance, increases the detection rate to 36%."

Line 501: Not only that, it is where we can be sure that there is enough signal in the data themselves to give a meaningful trend.

*This is correct – we have added this to the sentence.*

This is indeed a very interesting comment, and the main questions that popped into my mind when viewing the plots of the differences between trend and growth rate. TO be honest, I found the concept of these difference plots quite hard to grasp. How do you explain them? for e.g. what does it mean that for NO2 these differences can range around +/-5% per annum?

As discussed in Section 5.2 this is the difference between the linear trend (in %/yr) and the growth rate (in %/yr). In the text we discuss the difference between the trend and growth rate at Ibri (in Oman), we have now amend this

sentence to show explicitly what the difference implies. These differences between the linear trend and the growth rate occur due to the latter tracking inter-annual variations in the data not accounted for in the linear part of equation 1 [see Thoning et al. (1989) or https://www.esrl.noaa.gov/gmd/ccgg/mbl/crvfit/crvfit.html] – we have added this information to the text.

Lines 553, 561 & 570: Hence, you may conclude that for the DOMINO NO2 product, the estimated NO2 emissions will remain more or less unaltered between using a 10km or a 20km smoothing radius around the source. Do you agree? if so, add a relevant phrase in this paragraph.

The reviewer may be 'right' but we believe it is more accurate to just say that only the NO2 columns are generally unaltered.

Line 577: Yes, but these are not so great as to recommend that someone uses 10km instead of 20km. Please also recall that the usable OMI pixels are larger than 15x50km anyways.

Agreed, there is little to chose between the too values, although a 10 km radius much better covers the urban footprint of most locations, even the larger cities.

Line 621: So, you recommend using the affected rows?

Yes, but only to avoid trends in the number of samples, and, additionally only if the corrupted observations from those affected rows are heavily weighted so as not to affect the gridded data (i.e. by assigning each observation an extremely high uncertainty).

Line 630: You mean you didn't not use the IQR to exclude daily values from creating the monthly means? if you are worried about genuine points why don't you use 3\*IQR instead of 1.5\*IQR?

We do not understand the reviewer's comments in this instance. We do not filter the any of the satellite data using the IQR (nor is this indicated in the manuscript anywhere).

**Reviewer #2**

The paper contains a very thorough analysis of some air quality gases in the Middle East. It is scientifically speaking not very new, but it is a very useful overview of the air quality in the region with many details.

Line 160: Scenes with effective cloud fraction > 20 % are rejected for CHOCHO, in line 177-178: a filter for cloud radiance fraction > 0.3 is mentioned for SO2, while in line188 it is mentioned that scenes with a higher than 20% fractional cloud cover are filtered for all data. Why are you not using a single cloud filter setting for all data.

As mentioned in our response to reviewer 1, a cloud radiance fraction of 30% for SO2 corresponds to an effective cloud fraction of about 15-20%, making it consistent with the cloud filters applied to the other species. We have added this detail to the text, and adjusted the text accordingly.

Line 180: Several times in the period of this research volcanoes in North-East Africa have been erupting with as result volcanic plumes over the Middle East region. Large parts of the plume have values of less than 5DU. For example in June 2011 many days show remnant SO2 values caused by the eruption of the Nabro volcano. I would expect this affects the trend considerably and it might be better to remove this period from the time series of SO2. *We removed the June 2011 data and repeated our analysis – it had negligible effect – only producing 1 extra trend over a power plant. Note also that anomalous SO2 peaks potentially due to volcanic eruptions, are typically removed by the outlier filtering as discussed in Section 3.3*

Line 200-201: Does this mean that for the gases HCHO and NO2 the row anomaly mask in 2013 is less strict than in some earlier years? By using different criteria over the years for these gases your trend is still affected by sampling I would think. Please clarify this section.

We tested many different approaches to account for the row anomaly when gridding the data. The static mask, based on the most affected rows at the end of 2013, simply identifies the most problematic detector rows. However, those detector rows not covered by the static mask are also sometimes compromised for certain time periods during the mission, but they can (& do) provide valuable observations outside those periods. However, note each retrieval group assign different quality flags (QFs) to their product, which we also have to respect when gridding the data. For SO2 and CHOCHO the static mask method & product QFs work fine, but for NO2 and HCHO it doesn't. Hence this is why our approach for these latter gases is to block out the worst rows using the 2013 mask, and then assign a high uncertainty to any other row- corrupted observations. This compromise yields a high number of observations per grid cell and the lowest number of sample trends. The additional sensitivity test (test 3 in section 5.2) only uses data from rows in which there have been no corrupted observations over the lifetime of the mission.

Line 224: I think this Figure 2 does not add much to the paper and in my view can be removed.

*Figure 2 has now been moved to the supplementary material, as we believe it is important that readers can view how well the spatial mask captures the urban extent of target areas.*

Line 579: Also for NO2 a cloud fraction of less than 20% is advised. However, I doubt these advices were given with this particular test in mind. I suggest to apply this test also to CHOCHO and SO2.

As requested we also applied a stricter effective cloud fraction filter of 10% to all species - for SO2 this correspond to a cloud radiance fraction of 20%. This had little impact on SO2 although fewer trends in total detected for NO2 (232) and HCHO (59); no trends were found for CHOCHO. Generally, most of the highest ranked trends for each species were very similar to the default scenario, indicating a 20% cloud fraction filter is likely an optimum choice for detecting air-quality trends over this region without affecting trend magnitudes. This information has been added to the text.

Line 595-599: I thought the standard analysis was already done for unaffected rows. How are the unaffected rows defined in this particular test? *See our comments above.*

Line 622: It is not only the number of observations but also the type. Because you are using a different selection of rows you have a selection of other pixel sizes. The change of pixel sixe alone will already affect the derived trends. This should be added to the discussion.

We have added this interesting point to the discussion. Thanks!

Section 5.3: In this section it might be interesting to include the study of Schneider et al. (2015), who did a trend analysis on NO2 in large urban agglomerations for the period 2002-2012, based on SCIAMACHY. *Added.*

Figure 3, left-side: In this Figure I have difficulties distinguishing between oil refineries, oil ports or power plants. Other symbols or colors can improve the Figure. Please also add the symbols to the legend. *Done.*

Table 1-4: In my opinion large part of this table can be moved to the supplementary Material

We prefer to keep Tables 1-4 as they are, since they provide a summary of all the key statistics for each species (rather than having too many additional tables in the supplementary material – as this was commented on in the initial pre-ACPD review).

In general: the significance in which most values are given is much too high compared to their errors. A digit less is often possible and makes the text better readable.

We prefer to keep the level of significance (& decimal places) as originally presented, as it is more accurately helps to distinguish & rank the linear trends and pollution levels – and is consistent with the tabulated results in the excel files provided in the supplementary materials.

**Additional References**

Marais, E. A., Jacob, D. J., Kurosu, T. P., Chance, K., Murphy, J. G., Reeves, C., Mills, G., Casadio, S., Millet, D. B., Barkley, M. P., Paulot, F., and Mao, J.: Isoprene emissions in Africa inferred from OMI observations of formaldehyde columns, Atmos. Chem. Phys., 12, 6219-6235, doi:10.5194/acp-12-6219-2012, 2012.

MOP-KRG: Building Kurdistan Region of Iraq: The Socio-Economic Infrastructure, Ministry of Planning Kurdistan Regional Government, 2012.

Mustafa, Y. T., Ali, R. T., and Saleh, R. M.: Monitoring and Evaluating Land Cover Change in The Duhok City, Kurdistan Region-Iraq, by Using Remote Sensing and GIS, International Journal of Engineering Inventions, 945 1(11), 28–33, 2012.

Sabolis, A., Meskhidze, N., Curci, G., Palmer, P. I., and Gantt, B.: Interpreting elevated space-borne HCHO columns over the Mediterranean Sea using the OMI sensor, Atmos. Chem. Phys., 11, 12787-12798, doi:10.5194/acp-11-12787-2011, 2011.

Stammes, P., M. Sneep, J. F. de Haan, J. P. Veefkind, P. Wang, and P. F. Levelt (2008), Effective cloud fractions from the Ozone Monitoring Instrument: Theoretical framework and validation, J. Geophys. Res., 113, D16S38, doi:10.1029/2007JD008820

[revised manuscript text omitted]